# Position: AI for Just Work:
# Constructing Diverse Imaginations of AI beyond "Replacing Humans"

## Abstract

The AI community usually focuses on "how" to develop AI techniques, but lacks thorough open discussions on "why" we develop AI. Lacking critical reflections on the general visions and purposes of AI may make the community vulnerable to manipulation. In this position paper, we explore the "why" question of AI. We denote answers to the "why" question the imaginations of AI, which depict our general visions, frames, and mindsets for the prospects of AI. We identify that the prevailing vision in the AI community is largely a monoculture that emphasizes objectives such as replacing humans and improving productivity. Our critical examination of this mainstream imagination highlights its underpinning and potentially unjust assumptions. We then call to diversify our collective imaginations of AI, embedding ethical assumptions from the outset in the imaginations of AI. To facilitate the community's pursuit of diverse imaginations, we demonstrate one process for constructing a new imagination of "AI for just work," and showcase its application in the medical image synthesis task to make it more ethical. We hope this work will help the AI community to open dialogues with civil society on the visions and purposes of AI, and inspire more technical works and advocacy in pursuit of diverse and ethical imaginations to restore the value of AI for the public good.

## 1. The imagination of AI: What is AI for?

The state-of-the-art data-driven technologies, now primarily called artificial intelligence/machine learning (AI/ML) (Goodlad, 2023), have gained great momentum and societal attention in the past decade. In the AI community, we often ask "how" to propose, develop, and deploy AI techniques, but rarely reflect on the "why" question on the purposes of AI. Lacking critical reflections on the fundamental "why" question may render the community susceptible to being trapped by suboptimal objectives, influenced by distorted research agenda, and manipulated for unjust purposes (Greenhalgh, 2024; Brynjolfsson, 2022; Sismondo, 2018). In this work, we explore the "why" question in AI. We ask, "Why do we develop AI? What purposes does AI serve?" We denote answers to these inquiries the imaginations of AI, which depict the overarching motivations of AI, i.e., how AI can be helpful to humanity and the planet. Extended from the concept of imaginary in social science (Benjamin, 2024; Earle, 2020), the imagination of AI is the general vision, frame, and mindset we have for the prospects of AI technologies. The imagination of AI sets the agenda on what are important goals to pursue in AI that reflect our value systems, and shapes assumptions, design choices, and objectives when developing specific AI techniques.

Of course, different people have different imaginations of how AI can be helpful for humanity and the planet. As a community, our collective imaginations should also be diversified to reflect and accommodate a variety of pursuits from people with different backgrounds, values, and interests. While different visions of AI exist in the community, and such works usually appear in venues and subfields such as AI for (Social) Good, AI for Science, or the ICML Position Paper Track, they are either at the inception or in a relatively niche position regarding their overall influence[1]. In comparison, the long-established mainstream imaginations of AI in the community are dominated by a monoculture that is shaped by "a small and largely homogeneous group of people" (Burrell & Metcalf, 2024). This mainstream imagination of AI emphasizes progress and performance, encoding values such as performance, generalization, efficiency, and novelty according to a study on 100 highly-cited papers in ICML and NeurIPS (Birhane et al., 2022). The prominent manifestation of this imagination within the AI community

---

[1] There are more critiques with different visions of AI from other fields like critical theory, FAccT, STS, see Appendix B for details. Our work draws on these critiques and diagnoses, and contributes to the AI community by adapting them to the technical context to bring them into the community's focus, and bridging diagnoses with treatment in AI technical practice.

[1] Anonymous Institution, Anonymous City, Anonymous Region, Anonymous Country. Correspondence to: Anonymous Author <anon.email@domain.com>.

Preliminary work. Under review by the International Conference on Machine Learning (ICML). Do not distribute.

Figure 1. Roadmap of this paper that highlights the structure and key contents of each section.

and in the public is the AI objective to automate human occupations and to surpass and replace humans, evidenced by a recent large-scale survey with 2,778 AI researchers on the future of AI (Grace et al., 2024) and the widespread societal anxiety of being replaced by AI (Ivanov et al., 2020).

While the mainstream imagination has some merits, it is not without flaws. In this imagination, there is no room to encode values that can directly improve human and environmental welfare at the outset in the blueprint of AI, such as justice, democracy, and sustainability. Instead, these ethical values are regarded as secondary priority and mainly used as post-harm fix rather than prevention from the root. Furthermore, fixating on a single-minded imagination is a problem for the community, indicating other possible imaginations and the diverse values underpin them are suppressed (Marcus, 2024; Burrell & Metcalf, 2024; Rudin & Wagstaff, 2013). People who envision different imaginations could feel unwelcome and their contributions unrecognized, which is the partial reason for the current lack of diversity and inclusion in the AI community (Wajcman & Young, 2023).

To address these issues, in this work we call for 1) **diversified collective imaginations of AI** that 2) **synergize both ethical values and AI development** (Fig. 1). We first synthesize multidisciplinary evidence and rationales to understand why the current mainstream imagination itself is largely unethical and can cause harms (Section 2). We then highlight the important role of imaginations in technical and social actions (Section 3), and demonstrate a process of constructing a new imagination of "AI for just work" that bakes ethical values in its grounding assumptions and technical practice (Section 4). We hope this work can stimulate the AI community to open public debates and discussions with civil society on the "why" question of AI, especially with those who are vulnerable, marginalized, or negatively affected by AI, and support diverse imaginations, values, and works for the ethical pursuit of AI development, such that to restore the value of AI technology as a public good.

## 2. The mainstream imagination of AI (MIA) is not as ethical as we assume

### 2.1. Alternative views: MIA and its justifying statements

The mainstream imagination of AI (MIA) is self-explanatory and seldom questioned, probably because it is based on

fundamental assumptions that are the basic beliefs within the AI community and in society. The typical logic to justify that MIA is an ethical pursuit for social good can be summarized in the following statements (**S1**-**S4**)[2]:

**S1**. *As AI advances, it may converge into being able to solve very hard problems, such as surpassing human intelligence, solving scientific and social problems such as global warming and inequality, and fixing its own flaws and harms. Each solution has the potential to bring prosperity to humanity.*

**S2**. *As AI surpasses human intelligence, AI can be used to replace humans. Doing so can free human beings from toil.*

**S3**. *By replacing human labor with AI, it will also release great productivity and efficiency, which will lead to human prosperity.*

**S4**. *If harms occurred during AI progress, the harms are often caused by bad actors or intentions that maliciously develop or use AI.*

Given the critical role of these statements in legitimizing the ethical pursuits of MIA and their consequences in society, these statements cannot be taken at face value. Rather, the AI community has the responsibility to either critically examine the validity and ethics of these statements, or provide solid evidence and rationales to defend these statements in the face of counterarguments to establish their validity and ethics in supporting MIA. Since both aspects are currently missing in the community, next we conduct a critical examination on MIA and its justifying statements **S1**-**S4** by synthesizing evidence and rationales from philosophy of science (**S1**), ethics (**S2**), feminism (**S2**), economics (**S3**), and social science (**S4**).

### 2.2. Critical examination of MIA

**S1**. *AI can be omniscient.*

**Summary of our counterarguments**: S1 surreptitiously substitutes the model for the real, and ignores AI's fundamental dependencies on human experience and nature.

The imagination to create a God-like, super human intel-

---

[2]We note of course that even for the mainstream viewpoints in the AI community, there are substantial diversity and nuances in viewpoints that S1-S4 do not capture.

ligence that is capable of solving any hard problem for humanity at our will incarnates the unspoken monotheistic mindset underlying modern science and technology. Frank et al. (2024) name this mindset the blind spot worldview. It is characterized by the philosophical perspective that there exists a rational, "disembodied, God's-eye perspective" for us to gain "access to a perfectly knowable, timeless objective reality" (Frank et al., 2024). This worldview divorces the knower and the known, subjectivity and objectivity, forgets their mutual relationship, and unnecessarily considers objectivity to have a higher hierarchy than subjectivity regarding their closeness to the idealized, God's-eye view of truth (Fig. 2). Entangled with culture, politics, and economy, this limited perspective on science and technology was woven into the fabric of our value systems in society, which values rationality, abstraction, objectification, quantification, order, control, and downgrades experience, intuition, sense, emotion, fluidity, and ambiguity. It, in accordance, frames how we regard the human experience and the natural world, which relegates them to pure resources to control and exploit (Frank et al., 2024). As Frank et al. (2024) argue, this "impoverishes the living world and our experience" and leads to existential crises such as environmental destruction, global pandemic, and algorithmic surveillance. Next, we examine the particular costs and harms of MIA to human labor and society.

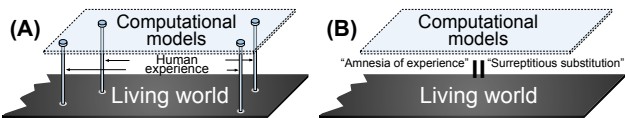

*Figure 2.* Visualization of two contrasting worldviews of science and technology including computational models. **(A) The normal worldview**: it acknowledges and values the fact that computational models are built upon human experience (embodied as labor, knowledge, dataset, infrastructure, etc.), and rely on it to validate and apply models in the real world; human experience and models are like pillars and floor in the figure: they are mutually dependent on each other to better understand our living world; computational models are useful reductive abstractions of, but cannot fully represent, the complex real-world phenomena they model (the well-known aphorism that "all models are wrong, but some are useful" (Box, 1976)). **(B) The blind spot worldview**: it wrongly assumes that computational models are dominant over human experience, "belies AI's fundamental dependence on physical nature and socially organized, collective human knowledge" ("the amnesia of experience"), "substitut[es] computation for genuine embodied intelligence," and "treat[s] abstract computational models as if they were concretely real" ("confuse the map with the territory"). Unspecified quotations in the figure and caption are from Frank et al. (2024).

**S2**. *Replacing human with AI can free human.*

S2 forgets AI's fundamental dependence on labor, devalues workers and degrades work to make AI appear "automatic."

The imagination to replace humans by AI forgets the mutually dependent relationship between computational models

and human experience (Fig. 2-A), and assumes that AI can be dominant over human experience, which is a manifestation of the blind spot worldview (Fig. 2-B). By forgetting humans' fundamental role as the "epistemic footing" (Frank et al., 2024) for AI, it 1) "wrongs someone in their capacity as a subject of knowledge, and thus in a capacity essential to human value," which manifests epistemic injustice in ethics (Fricker, 2007). It also 2) impoverishes our human experience of work that is supposed to be "meaningful, fulfilling activities" as the "social means necessary to live a flourishing life" (Wright, 2021). We expand the two interrelated harms below.

First, the imagination of labor replacement for human liberation is not new. Atanasoski & Vora (2019b) identify that the narrative of AI replacement shares similar patterns and assumptions with the long-standing labor replacement phenomena since the inception of capitalism: "the emergence of reproductive unpaid women's work in Europe [due to witch hunts]...replace[d] the rebellious peasants opposing the enclosure of common lands and resources;" In the US plantations, farmers and indentured servants were replaced by slaves; In globalization, domestic or in-house workers were replaced by offshore or outsourced workers; Nowadays in gig economy, full-time, permanent workers are replaced by precarious workers such as contract, part-time, or gig workers, who in turn generate data to train AI and are expected to be replaced by AI, a.k.a., the machine slaves[3] (Atanasoski & Vora, 2019b). In each labor replacement, the original workers are replaced by workers who are less paid, poorly treated, enjoy much less freedom in their work, are less likely to rebel, are atomized and isolated thus harder to unionize and have little to no bargaining power. In other words, labor replacements greatly reduce the power of workers who are replaced and who replace others; and the proposed human liberation from labor replacements is merely "enabling and improving the lives of privileged subjects" (Atanasoski & Vora, 2019b).

For such labor exploitation and oppression to happen, systematic violence (Federici, 2004), state and imperial forces (Mies, 1986), and divide and conquer strategy are needed to establish gendered, racialized, and other social hierarchies that legitimize the exploitation of the "degraded and devalued others—those who were never fully human" (Atanasoski & Vora, 2019b). The MIA to replace humans is no different: by assuming a new human-machine hierarchy, it still reproduces the same logic of racialized and gendered social hierarchies that continue to legitimize more hidden labor exploitation (Atanasoski & Vora, 2019b). Specifically, as we mentioned, there are two groups of workers who are prominently harmed by AI replacement: those

---

[3]The word "robot" is from Slavic linguistic root "rab," which means "slave" (Jordan, 2019).

whose jobs are taken by AI, and those who work behind or with AI, such as Uber drivers (Rosenblat, 2018), Amazon Mechanical Turk workers (Gray & Suri, 2019), and Amazon warehouse workers (MacGillis, 2021). The latter group suffers from more hidden harms and exploitation, because their work and contributions are often purposely made invisible and are attributed as the "intelligence" of the machine (Atanasoski & Vora, 2019b), reflecting the blind spot worldview and epistemic injustice. Since AI constantly depends on human labor and experience (Frank et al., 2024), as long as the hierarchical assumption in AI exists, these harms will persist no matter how AI advances.

Second, the MIA to replace humans not only intensified labor exploitation as described above, but also impoverished the concept of work. Ironically, in the discourse of AI replacing humans, the bogus "autonomous" aspect of machine is emphasized, but the real human autonomy, which is one of the five key principles in AI ethics (Floridi et al., 2018), is ignored. Consequently, workers and work tend to be organized around the capabilities and limitations of the machine, not the other way around (Delfanti, 2021). As Frank et al. (2024) note, one aspect of "the computational blind spot...is failing to see how we are led to remake the world so that it gears into the limitations of our computational systems." The deprived workers' power and autonomy reduced work to a single dimension of the necessary means (or drudgery) to earn one's living (Resnikoff, 2021), whereas the multifaceted meaning of work, such as a social means to live a meaningful and dignified life or connecting with people and the community, can only be fully enjoyed by a privileged few. The scope of work was also narrowed that defines whose and what kinds of activities count as work and can enjoy corresponding protection, support, and recognition, and what don't (Mies, 1986). The refashion of our living world and work to adapt to machine greatly limits everyone's living experience and options on how we can live a flourishing life in a originally meaning-rich world.

**S3**. *AI can promote productivity and prosperity.*
S3 is more likely to lead to inequality, as it inherently lacks power check mechanisms to ensure shared prosperity.

Although **S2** has many unreasonable aspects and obvious harms including the above counterarguments, a strong support for it is in **S3** that AI advances can raise productivity, which will eventually benefit us all despite the harms. However, economic studies have shown that such "lift all boats" effect does not happen automatically with technological progress, but is the result of both 1) our choices of how technology is developed and 2) active labor struggles and bargaining (Acemoglu & Johnson, 2023).

First, AI does not necessarily lead to productivity gain, and not any types of productivity gain can precondition the broad-based prosperity. Not all AI technologies increase productivity. For example, AI-enabled surveillance techniques are not mainly focused on improving productivity, but to impose more control over workers, reduce their power, and save on wage costs (Acemoglu & Johnson, 2023). Furthermore, replacing labor with AI tends to shift the production in favor of capital and against labor, as it creates a strong displacement effect that reduces labor demand and wages, which is in the opposite direction to broad-based prosperity. When the productivity gain from automation is not enough to offset the displacement effect, it is the so-so automation or excessive automation that lowers productivity growth and widens inequality (Acemoglu & Restrepo, 2019a;c). Empirical evidence in the US and Germany shows that exposure to industrial robots or automation reduced labor income share, which impacted low-skill workers more and thus exacerbated inequality (Acemoglu & Restrepo, 2022; 2020; Dauth et al., 2017). The displacement effect may be countervailed and labor share may be increased only when workers' marginal productivity is increased. Distinct from the standard definition of productivity (or average productivity), which is a worker's average output, marginal productivity is "the additional contribution that one more worker brings by increasing production or by serving more customers" (Acemoglu & Johnson, 2023). Marginal productivity comes from the increased productivity that is large enough to expand jobs and employment, from technologies that can aid workers, and more importantly, from technologies that can create new labor-intensive tasks "in which labor has a comparative advantage" (Acemoglu & Restrepo, 2019b;c). These directions, as Acemoglu & Johnson (2023) point out, are all highly dependent on the different visions we choose to develop AI.

Second, the increased marginal productivity alone is not enough for broad-based prosperity, but should be accompanied by strong labor power that guides the direction of technological development and ensures that marginal productivity is shared with workers (Acemoglu & Johnson, 2023). The increased demand for labor does not automatically increase workers' wages, because employer and employee are in a coercive relationship, the employer may not face external competitors for workers, and wages are determined not only by market forces but also by bargaining power (Acemoglu & Johnson, 2023). Acemoglu & Johnson (2023) review the economic history and technical change from the Middle Age to present, and associate the changes in workers' wages, welfare, working conditions, and directions of technical development with changes in social and political power.

**S4**. *Harms of AI are due to malicious use.*
S4 makes the wrong diagnosis.

Our examination of **S1**-**S3** shows that unethical issues in AI are rooted in intertwined causes encoded in the MIA includ-

ing: a limited philosophical worldview, unjust hierarchy in social structures of labor, and imbalanced economic and political power. Embedded in the social system, AI technology acts as a mediator, is shaped by these root causes and in turn shapes the unethical impacts of AI in society (Matthewman, 2011; Boenig-Liptsin et al., 2022). Tackling the malicious use to solve unethical issues in AI only treats the symptoms, not the disease. Similarly, the imagination that AI can fix its own issues in **S1** is not based on valid logic. Solving a problem requires first conducting a precise diagnosis of the root cause, and then identifying treatment that can target the diagnosis. There is no guarantee that the right treatment will always fall into the realm of the technical toolbox, and claiming AI can fix any issues is like selling a cure-all elixir (Narayanan & Kapoor, 2024).

## 3. Ethical and diverse imaginations are needed for AI

In the previous section, we critically examine the MIA and defamiliarize ordinary consensus that we take for granted in the community. People may defend MIA by the benefits brought by AI. Our examination does not reject the benefits either. The disagreement lies, is it really necessary to couple AI development with the harms? Or, is it possible to imagine different visions of developing AI that have the benefits and get rid of the harms? Our critical examination points out that the root causes of harms are not in technology itself, but lie in the flawed assumptions in the imagination that underpins technology. These flawed assumptions are not a necessary component in technological development; rather, they are highly dependent on our choices of how to shape our imagination. Technological development is not deterministic, and harms may not be an inevitable side effect of technology. This means that it is possible to decouple harms from AI development by getting rid of the unethical assumptions in the imaginations of AI, which is identical to replacing the unspoken and unjustified unethical assumptions with transparent and justified ethical assumptions from the outset in the imagination of AI.

However, it should be noted that the reason for MIA to be the dominant imagination is that it is deeply rooted in the dominant value systems and social and power structures that set the paths of least resistance (Johnson, 2014) for people to act and succeed without the efforts to critically examine "why" questions behind actions, such as setting the agenda of trendy research topics (Rolnick et al., 2024; Rudin & Wagstaff, 2013; Wagstaff, 2012), common design assumptions, popular benchmarks (Birhane & Prabhu, 2021; Koch et al., 2021; Raji et al., 2021), and shaping common practices in evaluation (Reinke et al., 2024; Jin et al., 2024), reporting (Burnell et al., 2023), and peer review (Andrews et al., 2024; Lipton & Steinhardt, 2019). As a consequence, this greatly limited our ability to imagine organizing our so-

cial relationships and our relationship with AI in alternative visions (Benjamin, 2024; Costanza-Chock, 2023). Choosing to deviate from MIA (the path of least resistance) means that individuals and the community need to constantly investigate efforts in collective imaginations to "get over some of this deeply habituated laziness and start engaging in interpretive (imaginative) labor for a very long time to make those realities stick" (Graeber, 2015). Despite the hard work, such endeavors are essential for the healthy development of AI and the research community, if we want to correct the current trend of AI research being captivated by private sectors' values and interests (Young et al., 2022; Birhane et al., 2022; Zheng et al., 2024), and restore the value of AI research for public good. That means the AI community needs to be aware of the problems in MIA by engaging in scientific criticisms from inside and outside the community and making the self-correcting mechanism in scientific research work for our community, and starts to recognize, support, and cultivate diversified imaginations of AI for various ethical pursuits and enable them to be on the paths of least resistance in the research processes, such as research topic identification, funding, and peer review.

## 4. AI for just work: a demonstration of constructing a new imagination

Diverse imaginations can be constructed by focusing on various ethical values such as social justice, environmental justice, the values of community and solidarity, etc. And specific technical works can build upon one or the combination of several ethical imaginations. To inspire and facilitate the community to construct diversified imaginations of AI, in this section, we demonstrate our process to construct a new imagination of "AI for just work" and apply it in our specific AI development. We first construct and justify three foundational assumptions that form the value system for the new imagination, and deduce new properties of AI from these assumptions. We then show the process of how the new imagination guides specific technical decisions in the case study of medical image synthesis task.

### 4.1. Foundational assumptions

Our imagination of "AI for just work" focuses on the relationship between work and AI, because work is the main human activity and the major area in which AI applications are involved. This new imagination is underpinned by three foundational assumptions (**A1**-**A3**) that reflect our ethical values in the worldview of AI and in work justice.

**A1**. *As computational models, AI models can be useful abstractions or simplifications of, but cannot fully represent, the complex phenomena they model; AI models are grounded in human experience and are dependent on the material world.*

We define complex phenomenon by borrowing the definition

of complex systems, which are "co-evolving multilayer networks," are context-dependent, and are composed of many non-linearly interacting elements (Thurner et al., 2018), such as language, human behavior, human mind, a living cell, financial market, social or natural phenomena. **A1** is justified by our counterarguments to **S1** and the normal worldview of science and technology (Fig. 2-A). This worldview acknowledges the limits of science and technology that science "isn't a window onto a disembodied, God's-eye perspective. It doesn't grant us access to a perfectly knowable, timeless objective reality. ...Instead, all science is always our science, profoundly and irreducibly human, an expression of how we experience and interact with the world" (Frank et al., 2024). Similarly, **A1** assumes that there does not exist a disembodied, God-like intelligence of knowing that grants us access to a perfect world model that can replace the original phenomena. It acknowledges and respects the fact that AI models are built upon human experience (embodied as human labor, prior knowledge, dataset, infrastructure, etc.), natural resources and materials of our planet (Crawford, 2021). This assumption is also well-supported by common framing in the academic community interested in machine learning and statistics. However, it stands in contrast to how AI is sometimes described in the tech industry.

**A2**. *Workers and their work are not devalued or degraded.*

**A2** is justified by our counterarguments to **S2** that the root causes of AI-based algorithmic oppression and exploitation are in the social hierarchy that wrongly assumes some people and their work are inferior, and in the imbalanced power between the oppressor and the oppressed. To correct the unjust assumptions, **A2** assumes that workers and their work are not devalued or degraded by any means, including but not limited to: workers' age, gender, sexuality, race, ethnicity, religion, nationality, physical or mental conditions, educational level, socioeconomic status, whether the work is paid or unpaid, the type, role, and salary of the work, the amount of efforts or skills required or engaged, how workers are rated and their ratings, etc. Furthermore, in this new imagination, we refer to critical theory and adopt a broader definition of work to revive its original meaning as "meaningful, fulfilling activities" as the "social means necessary to live a flourishing life" (Wright, 2021). Work under this assumption includes a wider range of human activities that may not count as "work" in the current narrowly framed economic system based on market exchange, such as reproductive labor and emotional labor (Raworth, 2017; Mies, 1986). In the context of AI, workers include a wide range of people who work for or behind AI (such as data labor), who work with AI (such as users), and whose work or life is impacted by AI (such as stakeholders). Using the broad concept of work ensures that people and their visible or invisible labor and contributions can be recognized, supported, and deserve workers' power to protect

their rights. In the remaining parts of the paper, we use the word "workers" to denote the broad groups of people that include workers, users, stakeholders, and the public. The ideas in **A2** are also justified by AI ethics principles of autonomy and justice (Floridi et al., 2018).

**A3**. *The values of work and other essential human, scientific, social, and environmental values are prioritized over growth, productivity, and efficiency.*

**A3** is justified by our counterarguments to **S3** that merely pursuing productivity and efficiency cannot automatically lead to shared prosperity, as this goal can easily be hijacked by the dominant power to serve the benefits and interests of the powerful. Instead of pursuing progress and productivity at the expenses of essential ethical values, it is more reasonable to maintain dynamic balance between ethics and development for the same goal of the mutual flourishing of humanity and the planet. The ideas in **A3** are also justified by AI ethics principles of beneficence and nonmaleficence (Floridi et al., 2018).

### 4.2. AI properties

The role of **A1**-**A3** to the new imagination is like axioms to a new axiomatic system. Next, we try to deduce some properties (**P1**-**P4**) that AI has under this new imagination.

**P1**. *The problem formulation, design, and evaluation of AI are grounded in real-world tasks and applications.*

As shown in **A1** and Fig. 2, the blind spot worldview (Frank et al., 2024) and the abstraction nature of AI models tend to get people caught up in the upper abstraction space, ignoring the grounded real-world space where AI models are applied and can have an impact. As Rolnick et al. (2024) point out, "ML algorithms designed in blue-sky, methods-focused research continue to fall short when used directly for applications." According to **A1**, because AI models abstract and simplify real-world phenomena, the modeling process creates inevitable information loss due to the gap between concrete world and abstract model. Therefore, considerable human efforts are indispensable (visualized as pillars in Fig. 2-A) to mitigate the gap and information loss. Human effort acts in a two-way process from real world to AI model and back to real world: First, in the modeling process, human labor is needed to determine which information from the real world is relevant for AI abstraction, such as the efforts in problem formulation, data curation, data labeling, and model design; Second, to apply AI in the real world, human labor is necessary to act as a buffer between AI model and real-world tasks to deal with the limitations of model and the ambiguity and fluidity of the real world, such as efforts in model evaluation, reorganizing the existing workflow, incorporating AI outputs in real-world tasks, long-term model monitoring and quality control. The current main paradigm of methods-focused AI research is not

driven by and grounded in the real world problems, making the above indispensable human/social processes not being seriously considered and incorporated in the modeling process. This greatly limits the positive impacts of AI research to society and impedes AI research itself (Rolnick et al., 2024). Furthermore, failing to ground AI models in real world applications tends to undervalue human labor in AI development and deployment, and underestimate the negative impacts of AI implementation on workers, which violates **A2**. **P1** joins the recurring call in the AI community for real world problem-grounded AI (Rolnick et al., 2024; Rudin & Wagstaff, 2013; Wagstaff, 2012), and provides additional justifications for this call based on the new imagination.

To ground the problem formulation, design, and evaluation of AI in real-world tasks, the first step is to understand and acknowledge the inevitable gap between AI modeling in the abstracted space and its application in the real world, and should not conflate the model's capability at the abstracted level (confined by many idealized assumptions) with the model's real-world utility and performance, which manifests the pathology of "confusing the map with the territory" in the blind spot worldview (Frank et al., 2024). The second step is trying to mitigate the gap by relaxing the too idealistic assumptions confined by the toy problem formulation and benchmarks, and incorporating more domain- and task-specific knowledge and human experience in the modeling process. As Andrews et al. (2024) state, this may improve scientific validity and avoid unethical pitfalls for AI. For example, instead of framing an evaluation problem as AI vs. doctor on a benchmark dataset, which is unrealistic in clinical scenario, the ultimate evaluation of a medical AI should assess how the AI and clinical workers perform when AI is embedded in clinical contexts (Cabitza et al., 2017).

**P2**. *Thorough limitation analysis is a default in AI evaluation.*

The current evaluation paradigm of AI is lopsided as it mainly evaluates the positive aspects of the proposed AI technique, such as performance improvement and efficacy of a proposed function from ablation study, but the negative aspects of the proposed AI technique, such as its limitations and negative impacts, are at most briefly mentioned without an equivalent amount of thorough analysis as the positive aspects (Herrmann et al., 2024). The biased evaluation paradigm is driven by and reinforces the MIA that technical progress of AI is to reach an idealized image of AI that is perfect with no weaknesses; and technical progress is prioritized and harms can be fixed later (**S1**), which violates **A3** in the new imagination.

In the new imagination, we acknowledge that there does not exist a perfect state of AI, and AI always has its limitations and weaknesses no matter how technology progresses (**A1**). To avoid weakening workers' power from the in-

complete evaluation that only emphasizes AI strengths and human weaknesses but neglects human strengths and AI weaknesses (**A2**), and to avoid developing AI for the sake of technical progress and ignoring its negative aspects (**A3**), we propose that a thorough limitation analysis of AI should be a routine in AI evaluation. The limitation analysis includes the following aspects: 1) quantitatively and/or qualitatively assessing the scope, weaknesses, and failure modes, and associating potential social consequences with the limitations, so that users can have a holistic understanding of the strengths and risks of the proposed AI technique. 2) Being transparent and acknowledging the design-specific assumptions and limitations in problem formulation, model design, and evaluation methodology that may be mitigated in future work. 3) Acknowledging the intrinsic limits of the proposed AI technique that cannot be overcome with technological progresses. Setting up the thorough limitation analysis as an AI evaluation standard also aligns with the scientific practice in biomedicine or engineering that equally evaluate the scope, efficacy, side effects, and safety issues of a medication or an engineering product.

**P3**. *AI and data collection are rejectable.*

Because AI models and applications have scopes, limitations, risks, costs, and uncertainty (**A1**, **P2**), it is reasonable to deduce that AI may not be applicable in every scenario, thus AI and its data collections can be rejected. As Benjamin (2016) states, informed refusal is the corollary of informed consent in ethics. Refusal also corresponds to the right to withdraw in research ethics, and has already been implemented in laws. For example, GDPR (2018) states "data subject shall have the right not to be subject to a decision based solely on automated processing," and have the right to erasure data ("right to be forgotten"). These measures are meant to protect rights and shift power dynamics in favor of the powerless, in this innately imbalanced power structure where technical sectors and institutions are the dominant.

However, this refusal assumption is currently not embedded in AI technical development, which creates a strong framing effect that sets the given AI model and technical solutions as default and their necessity is rarely interrogated or challenged. As we mentioned previously in **S4**, not all problems are suitable to be solved by AI. Maybe the "problem" is not a real problem at all that needs to be solved, or maybe the root causes of the problem can be tackled by non-technical or low-technical approaches. Failing to recognize this framing effect falls into the "solutionism trap" (Selbst et al., 2019). Encoding refusal as the basic property of AI can avoid the trap of regarding AI and its data collection as the default and inevitable option. Furthermore, refusal is not the end goal, but a way of constructing a reciprocal relationship between the powerless and the powerful, such that the design of AI and data collection can incorporate

goals and perspectives of workers, especially the vulnerable and marginalized, at the very start (Benjamin, 2016; Zong & Matias, 2024), which is in line with **A2**, **A3**, and **P1**.

**P4**. *The design of AI should try to improve worker rights and power.*

Because certain AI design choices, such as automation, can create a displacement effect that reduces workers' wages and weakens their power (Acemoglu & Restrepo, 2019a;c), which devalues workers' work thus violates **A2**, an ethical and prudent design of AI should take the impact of AI on worker rights and power into consideration, and choose to design AI that could improve, rather than weaken workers' value in work. As we mentioned in the counterarguments to **S3**, such design choices can include AI that aids workers, AI that creates new labor-intensive tasks and opportunities in which workers have comparative advantages over machine (Acemoglu et al., 2023). In cases when AI displacement effect cannot be completely offset, mechanisms should be placed to compensate for the reduced workers' rights and power, such as retraining workers for new skills. Scholars also point out that the so-called "collaborative AI," "human-augmenting AI," or "human-centered AI" may not necessarily lead to increased worker power, as some may still reproduce a hierarchical and exploitative relationship that shifts the oppression to the less powerful (Atanasoski & Vora, 2019a; Costanza-Chock, 2023). Therefore, whether or not an AI design reduces worker power should not be determined by the powerful party of the technical sector or the institution who implements the AI, but should undergo democratic processes by incorporating a wide range of workers' voices and perspectives in AI design, especially the vulnerable and marginalized. Ultimately, this collective design paradigm in AI development depends on power check mechanisms and structural changes that shift power dynamics from capital to workers and tackle the root causes in social institutions that exploit workers and make work miserable, such as collective bargaining and legislation to protect worker rights and power (for instance, their rights to reject AI in **P3**), and changing the ownership of production from private-owned to public- or worker-owned. The AI community has the agency to propose technical works and non-technical initiatives to facilitate, rather than impede, such power checks and structural changes.

### 4.3. Applying the imagination: a case study

The foundational assumptions and AI properties can be applied and extended in different contexts and subfields of AI. Next, we showcase an example that demonstrates the process of applying the new imagination of AI in research and potential applications of the medical image synthesis (MISyn) task. The MISyn task is to generate "visually realistic and quantitatively accurate images" in biomedicine (Frangi et al., 2018). Our recent work

"Ethical Medical Image Synthesis" (anonymous reference, anonymous paper in the supplementary file) conducts a thorough ethical analysis and proposes ethical criteria for MISyn that align with the new imagination of "AI for just work." We highlight key points that link the imagination to corresponding actions in MISyn research and peer review.

Based on **A1**, we identify the intrinsic limits of MISyn, including that synthetic images are not automatically grounded in real medical phenomena unlike medical images. Ignoring this limit permits the misinformation of MISyn that could cause harms to clinical workers and patients. To prevent the misinformation of MISyn, we propose several ethical criteria, including setting up technical standards, terminology, and usage declaration to clearly differentiate synthetic images from medical images in all outputs of MISyn techniques.

Based on **P1** and **A3**, we propose the ethical criterion that the problem formulation of MISyn should ground in real-world medical problems. Otherwise, as revealed by our ethical analysis, it is easy to mistake the technical progress in the model space as the real progress in clinical problems, which only benefits technical sectors rather than non-technical stakeholders. We also propose the five-phased evaluation paradigm to ground the evaluation of MISyn in clinical settings.

The ethical criteria also emphasize the importance of conducting limitation analyses, and extend the three aspects of limitation assessment in **P2** to tailor to the risks and limits of MISyn techniques. The right to reject AI (**P3**) and the protection of stakeholders' rights (**A2**, **P4**) are embedded in our proposed non-technical checklist for stakeholders to question the appropriateness of MISyn. To facilitate the implementation of the ethical criteria in MISyn research and applications, we conduct paper reviews to critically analyze two MISyn works published in high-impact journals using the ethical criteria, and show concrete examples of the gap between existing practice and the ethical criteria guided by different imaginations. The detailed changes of practice are listed in Appendix A.

## 5. Conclusion

In this work, we explore the "why" question in AI. We point out flaws in the current mainstream imagination of AI, and demonstrate one possible way to reimagine AI for the flourishing of humanity and the planet. This work calls the community to critically examine and diversify our mindsets on the foundational assumptions of AI that are usually taken for granted. We hope this work can inspire more works to diversify our collective imaginations of AI for ethical purposes, and encourage collective efforts within and outside the community to create space and support for these diverse imaginations to grow and thrive.

## Impact Statement

This work originates from our concern about the increasingly negative impacts of AI to society and the environment. In this work, we try to provide our diagnosis and treatment of the root causes of negative social impacts of AI by tackling the imagination of AI. While we call for positive changes in our imaginations of AI and hope this work can create positive impacts to the AI community, we are also aware that good intentions or imaginations alone do not automatically guarantee good impacts, a prominent example is the "ethics washing" phenomenon in AI (Rességuier & Rodrigues, 2020; Wagner, 2019; Ochigame, 2022). Therefore, we embed ethical values in the methodology and technical practice of AI, and combine it with power check mechanisms that strengthen the social and state power to check technical power, such as conducting critical reflections of our practice to reflect our responsibilities as researchers, scientists, and citizens, opening dialogues with the public, protecting workers' rights to decide, and regulations. We hope implementation of these mechanisms can equip ethics with its teeth and steer towards more positive impacts of the diverse and ethical imaginations of AI to society.

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

# Appendix

## A. Applying the imagination of "AI for just work": Case study of the medical image synthesis task

In Section 4.3, we showcase our application of the imagination of "AI for just work" to the specific medical image synthesis (MISyn) task based on our recent work "Ethical Medical Image Synthesis" (anonymized reference). We use this case because this is our first and currently the only project that applies the new imagination in research practice. To supplement this case, we detail the changes of practice before and after applying the imagination in Table 1. In the "Ethical Medical Image Synthesis" paper, to provide a hands-on guidance on how to apply the proposed ethical criteria in research, we conducted paper reviews of two MISyn works published in high-impact journals in the medical image analysis field. The examples of practice in Table 1 are from the paper reviews.

The case study on the medical image synthesis task also indicates that even if the task is intended for good purposes, such as this task of AI for medicine, and even if the target domain of application has already established strong ethics in its practice, such as this target domain of medicine, the penetration of the mainstream imagination of AI (MIA) into technical practice can still pose unscientific and unethical risks (Zheng et al., 2024).

Table 1: Six pairs of change of practice: from the current practice guided by the MIA, to the proposed practice guided by the new imagination of "AI for just work."

| Practice | Description of the practice and its example | Underlying assumptions of the practice | Corresponding imagination of AI reflected by the practice |
|---|---|---|---|
| 1.**Current** risk prevention | There is a lack of technical awareness and standards to explicitly label synthetic images as *synthetic* to avoid the misinformation risk of MISyn. | It assumes that either synthetic images barely pose risks, or it assumes that the burden of risks of MISyn is not important, can be shifted from technical sectors to users, and can be fixed later. | It reflects the MIA that pursuing technical advances is a merit on its own, can be prioritized over accountability and other ethical considerations, and regarding risks as secondary for post-harm fix. |
| 1.**Proposed** risk prevention | We identify and analyze the misinformation risk of MISyn, and propose to set up specific technical standards, terminology, and usage declarations to prevent misinformation by labelling and declaring the use of synthetic images. | It assumes that technical progresses are not free but come with costs and risks, and such risks are better prevented in the beginning of technical development rather than post-harm fix. | It reflects the new imagination that AI has intrinsic limits (**A1**), and ethics is prioritized over technical progresses (**A3**). |
| 2.**Current** power to reject AI | The current MISyn development does not raise any possibilities that allow users to question the appropriateness of MISyn and refuse it. | It assumes that AI implementation and usage are always a default and can rarely be questioned. | It reflects the MIA that ignores workers' autonomy and rights to decide whether to implement or use AI or not (**S2**). |
| 2.**Proposed** power to reject AI | We develop a checklist for non-technical stakeholders, regulators, and the public to check if a MISyn fulfills the ethical criteria, and enable them to question the appropriateness of the MISyn. | It assumes that workers have the right to refuse AI (**P3**). | It reflects the new imagination that the design of AI should try to enhance worker power (**A2**, **P4**). |

*Continued on the next page*

Table 1: Six pairs of change of practice: from the current practice guided by the MIA, to the proposed practice guided by the new imagination of "AI for just work."

| Practice | Description of the practice and its example | Underlying assumptions of the practice | Corresponding imagination of AI reflected by the practice |
|---|---|---|---|
| 3.**Current** problem formulation | In the two analyzed MISyn works, the unique technical features of the proposed MISyn techniques are not based on how such unique features were motivated from and can be potentially helpful for clinical problems. Rather, the proposed MISyn are motivated by technical needs, such as the need for large-scale dataset, and technical advances and novelty, such as the new text-to-image generative AI techniques. | It assumes that technical advances are always good to pursue. Or it assumes that the abstracted technical problem formulation can directly translate to the concrete, complex, real-world problem, or the proposed one-technique-fit-all solution can automatically apply to different clinical problems. | It reflects the MIA that prioritizes technical progresses, surreptitiously substitutes the abstract problem formulation for the real-world problem (**S1**), and ignores and devalues the difficulty and human labor in adapting a technique proposed for non-specific purposes to specific problems (**S2**). |
| 3.**Proposed** problem formulation | We provide ethical criteria to ground MISyn techniques in clinical needs, and provide specific suggestions for the two MISyn works, such as grounding the need for synthetic images in clinical problems of wound segmentation when real data is extremely limited, and grounding the text-to-image generative feature in surgical simulation with user-customized text inputs. | It assumes that the adaptation of an AI technique to a clinical problem does not happen automatically and requires considerable human efforts. Therefore, it is the technical sector's responsibility to incorporate task-specific requirements into the technique design from the beginning. | It reflects the new imagination to ground the technical problem formulation in real-world tasks (**P1**). |
| 4.**Current** evaluation reporting standard | The two examined MISyn works conducted evaluations that may not follow rigorous scientific standards, including claims of benefits that were not backed up by evaluation results, claims of significant improvement in performance that were not backed up by statistical tests, imbalanced claims that were biased towards highlighting the good performance while downplaying the bad performance, and the use of private data and the conducting of user study that were not clear if they had obtained approval from research ethics board. | It assumes that the pursuit for technical advances is prioritized over the pursuit for scientific validity, rigor, and ethics in evaluation and reporting. | It reflects the MIA that prioritizes technical progress and performance over scientific and ethical validity. |
| 4.**Proposed** evaluation reporting standard | We provide the ethical criteria for technical and non-technical reviewers to check if the evaluation contains unscientific claims or overclaims. | It assumes that the AI evaluation and reporting should maintain high standards of scientific validity and rigor as in other research fields. | It reflects the new imagination that prioritizes ethical and scientific values over technical progress (**A3**). |

Table 1: Six pairs of change of practice: from the current practice guided by the MIA, to the proposed practice guided by the new imagination of "AI for just work."

| Practice | Description of the practice and its example | Underlying assumptions of the practice | Corresponding imagination of AI reflected by the practice |
|---|---|---|---|
| 5.**Current** evaluation scope | The existing evaluation paradigm of MISyn is mainly algorithmic evaluation grounded in the model space. It lacks technical standards to declare the limitations of such an evaluation scope and to differentiate it from the evaluation scope in the real-world problem space. This could make the public to mistake the technical progress at the model space as the progress in the real-world problem space[4]. | It assumes that technical advances in the model space can be extrapolated to the technical advances in the real-world problem space. | It reflects the MIA that surreptitiously substitutes the algorithmic advances in the abstracted space for the genuine technical advances in the real world (**S1**). |
| 5.**Proposed** evaluation scope | We propose a five-phased evaluation paradigm to differentiate the primary phase of algorithmic evaluation in the model space from the more advanced phases of evaluation in the real-world problem space. | It assumes that the technical advances in the model space cannot be generalized to its performance in the real world. | It reflects the new imagination that ground the ultimate technical evaluation in real-world problems (**P1**). |
| 6.**Current** limitation analysis | The two analyzed MISyn works only declare limitations in technical design, but do not document other aspects of limitations, such as the intrinsic limits of MISyn, and do not conduct thorough analysis to assess the potential side effects of the proposed MISyn. | It assumes that AI progresses have no intrinsic bounds or limits, and the current technical limitations can always be overcome with more technical progress. Or it assumes that the limitation analysis is not as important as the benefit analysis of AI to be identified, assessed, and reported extensively[5]. | It reflects the MIA that as AI advances, it can solve any problems, including its own limitations and weaknesses (**S1**). |
| 6.**Proposed** limitation analysis | We propose the ethical criteria of three aspects of limitation analysis, including the thorough limitation assessment quantitatively and/or qualitatively on the scope, weaknesses, and failure modes of the proposed MISyn technique, acknowledging and declaring the limitations in technical design and assumptions, and acknowledging the intrinsic limits of MISyn techniques that cannot be overcome by technical advances. | It assumes that AI has limits and weaknesses, and the role of thorough limitation analysis is equivalent to or even more important than the benefit analysis of AI. | It reflects the new imagination that AI technologies have intrinsic limits and fundamental dependencies (**A1**). |

[4]If we use the progress in biomedicine as an analogy for technological progress, then the progress from algorithmic evaluation in the model space is like biomedical progress in cells or mice, and the technical progress in the real-world problem space is like biomedical progress in patients. Because we cannot conclude that a new drug that is effective on cells or mice will also be effective in patients, similarly, we cannot conclude that a new AI technique that shows technical progress in the model space will also show technical progress in the real-world problem space. Moreover, if the problem formulation is grounded more in the real-world problem space, the abstracted problem in the model space is likely to share more similar structures with the real-world problem, then it may increase the likelihood of successfully translating the evaluation performance of the proposed technique from the model space to the real-world space.

[5]Even if the researchers themselves know the limitations, if not assessed and reported explicitly and extensively at the same level as the benefit analysis, because the report will probably be released in the public domain, it can mislead the public to think that for the reported AI technique, its benefits greatly outweigh its limitations.

# B. Bibliographic essay on the multidisciplinary knowledge background

Since the multidisciplinary references in the paper may be distant from the target readers of this paper in the technical community, to help readers gain a better understanding of the relevant background, we provide a bibliographic essay to extend background information from the referred disciplines, and to introduce additional relevant references on topics discussed in the paper. The main related disciplines are social science (including critical theory, Science, Technology, and Society (STS)), philosophy of science, feminism, complex systems, ethics, economics, and Fairness, Accountability, and Transparency (FAccT). The essay is organized by topics according to their order of appearance in each section. For all quotations, emphasis is in original.

## B.1. Section 1. The imagination of AI: What is AI for?

### B.1.1. THE CONCEPT OF IMAGINATION IN SOCIAL SCIENCE

Imagination, or social imagery, is a concept usually used in social science to describe "the set of values, institutions, laws, and symbols through which people imagine their social whole. It is common to the members of a particular social group and the corresponding society."[6] It is "the web of meanings that binds a particular society together" (Earle, 2020) and provides "structures that guide our collective and individual actions and values."[7]

In her book *Imagination: A Manifesto* (Benjamin, 2024), sociologist Ruha Benjamin outlines the role of collective imagination in social change, and she provides the definition of imagination in the Preface of the book as follows:

> "I go back and forth between imagination and imaginaries — conceptual kin, related but not identical. Although a bit jargony as a noun, an imaginary refers to collective projections of a desirable and feasible future. I find myself invoking imaginaries when I want to cast a critical light on the imposition of a dominant imagination that presents itself as appealing and universal. You'll see, too, that I refer to imagination interchangeably with dreams and dreaming, ideas and ideologies."

In her keynote speech at ICLR 2020 *Reimagining the Default Settings of Technology & Society*[8], Ruha Benjamin emphasizes the role of imagination as a form of domination as well as a way for social change:

> "Social norms, values and structures all exist prior to any given tech development, so it's not simply the impact of technology we should be concerned about, but the social inputs that make some inventions appear inevitable and desirable, which leads to a third provocation, that imagination is a contested field of action, not an ephemeral afterthought that we have the luxury to dismiss or romanticize, but a resource, a battleground, an input and output of technology and social order. In fact, we should acknowledge that most people are forced to live inside someone else's imagination, and one of the things we have to come to grips with is how the nightmares that many people are forced to endure are the underside of elite fantasies about efficiency, profit, safety and social control. Racism among other axes of domination helps to produce this fragmented imagination. So we have misery for some, monopoly for others. This means that for those of us who want to construct a different social reality, one that grounded in justice and joy, we can't only critique the underside, but we also have to wrestle with the deep investments, the desire even that many people have for social domination."

Our proposal of focusing on imagination to the path for social and technical changes is influenced by these schools of thoughts in this subsection on the thesis that imagination can shape and guide practice; and our collective agency is reflected in our capacity to reflect and alter our imaginations and make changes in reality accordingly. For example, in Chapter 1 *Dead Zones of the Imagination: An Essay on Structural Stupidity* of the book *The Utopia of Rules: On Technology, Stupidity, and the Secret Joys of Bureaucracy* (Graeber, 2015), anthropologist David Graeber describes the nature of imagination and the possibility to "give power to the imagination":

> "From a left perspective, then, the hidden reality of human life is the fact that the world doesn't just happen. It isn't a natural fact, even though we tend to treat it as if it is — it exists because we all collectively produce it. We imagine things we'd like and then we bring them into being.

---

[6] https://en.wikipedia.org/wiki/Imaginary_(sociology)
[7] https://socialimaginaries.org/the-imaginary-system-of-society/
[8] https://iclr.cc/virtual_2020/speaker_3.html

......

[T]he kind of imagination I have been developing in this essay is much closer to the old, immanent, conception. Critically, it is in no sense static and free-floating, but entirely caught up in projects of action that aim to have real effects on the material world, and as such, always changing and adapting.

......

If one resists the reality effect created by pervasive structural violence — the way that bureaucratic regulations seem to disappear into the very mass and solidity of the large heavy objects around us, the buildings, vehicles, large concrete structures, making a world regulated by such principles seem natural and inevitable, and anything else a dreamy fantasy — it *is* possible to give power to the imagination. But it also requires an enormous amount of work.

Power makes you lazy. Insofar as our earlier theoretical discussion of structural violence revealed anything, it was this: that while those in situations of power and privilege often feel it as a terrible burden of responsibility, in most ways, most of the time, power is all about what you *don't* have to worry about, *don't* have to know about, and *don't* have to do. Bureaucracies can democratize this sort of power, at least to an extent, but they can't get rid of it. It becomes forms of institutionalized laziness. Revolutionary change may involve the exhilaration of throwing off imaginative shackles, of suddenly realizing that impossible things are not impossible at all, but it also means most people will have to get over some of this deeply habituated laziness and start engaging in interpretive (imaginative) labor for a very long time to make those realities stick."

In his book *Envisioning Real Utopias*, sociologist Erik Olin Wright proposes the concept of emancipatory social science that "seeks to generate scientific knowledge relevant to the collective project of challenging various forms of human oppression" (Wright, 2010), and provides pathways to ground the imaginations for social change in pragmatic knowledge and understanding about the complexity of social systems and humanity. This work inspires us to base the new imaginations on "diagnosis and critique of the causal processes that generate these harm" (Wright, 2010). The following is an excerpt in Chapter 1 *Introduction: Why real utopias?* on the relationship between imagination and practice:

'The idea of "real utopias" embraces this tension between dreams and practice. It is grounded in the belief that what is pragmatically possible is not fixed independently of our imaginations, but is itself shaped by our visions. Self-fulfilling prophecies are powerful forces in history, and while it may be naively optimistic to say "where there is a will there is a way," it is certainly true that without a "will" many "ways" become impossible. Nurturing clear-sighted understandings of what it would take to create social institutions free of oppression is part of creating a political will for radical social changes to reduce oppression. A vital belief in a utopian ideal may be necessary to motivate people to set off on the journey from the status quo in the first place, even though the likely actual destination may fall short of the utopian ideal. Yet, vague utopian fantasies may lead us astray, encouraging us to embark on trips that have no real destinations at all, or, worse still, which lead us towards some unforeseen abyss. Along with "where there is a will there is a way," the human struggle for emancipation confronts "the road to hell is paved with good intentions." What we need, then, is "real utopias": utopian ideals that are grounded in the real potentials of humanity, utopian destinations that have accessible waystations, utopian designs of institutions that can inform our practical tasks of navigating a world of imperfect conditions for social change.'

In addition to our proposed new imagination, another example of the new imagination of AI is the Indigenous AI project. Named Abundant Intelligences, this new research agenda of AI is based on Indigenous knowledge systems and relational ethics. It reimagines how AI technologies can flourish the Indigenous communities and how AI development can be guided towards a more humane future (Lewis et al., 2024; 2020).

B.1.2. THE MAINSTREAM IMAGINATION OF AI

**1. The phenomenon of monoculture in the AI community, and why questioning the objectives of AI matters?**

Our inquiry of the imagination of AI echoes with previous rare but important works in the AI community to zoom out and critically inspect the objectives of AI. For example, in his speech *Not without us* (Weizenbaum, 1986), Joseph Weizenbaum encourages computer scientists and AI experts to critically reflect the purposes of our work:

'It is a prosaic truth that none of the weapon systems which today threaten murder on a genocidal scale, and whose design, manufacture and sale condemns countless people, especially children, to poverty and starvation, that none of these devices could be developed without the earnest, even enthusiastic, cooperation of computer professionals. It cannot go on without us! Without us the arms race, especially the qualitative arms race, could not advance another step.

......

In this context, Artificial Intelligence (AI) comes especially to mind. Many of the technical tasks and problems in this subdiscipline of computer science stimulate the imagination and creativity of technically oriented workers particularly strongly. Goals like making a thinking being out of the computer, giving the computer the ability to understand spoken language, making it possible for the computer to see, goals like these offer nearly irresistible temptations to those among us who have not fully sublimated our playful sandbox fantasies or who mean to satisfy our delusions of omnipotence on the computer stage, i.e., in terms of computer systems. Such tasks are extraordinarily demanding and interesting. Robert Oppenheimer called them sweet. Besides, research projects in these areas are generously funded. The required monies usually come out of the coffers of the military - at least in America.

It is enormously tempting and, especially in Artificial Intelligence work, seductively simple to lose or hide oneself in details, in subproblems and their subproblems, and so on. The actual problems on which one works - and which are so generously supported - are disguised and transformed until their representations are more fables, harmless, innocent, lovely fairy tales.

......

I don't quite know whether it is especially computer science or its subdiscipline Artificial Intelligence that has such an enormous affection for euphemism. We speak so spectacularly and so readily of computer systems that understand, that see, decide, make judgments, and so on, without ourselves recognizing our own superficiality and immeasurable naivete with respect to these concepts. And, in the process of so speaking, we anesthetise our ability to evaluate the quality of our work and, what is more important, to identify and become conscious of its end use.

......

One can't escape this state without asking, again and again: "What do I actually do? What is the final application and use of the products of my work?" and ultimately, "am I content or ashamed to have contributed to this use?" '

Kiri L. Wagstaf in her paper *Machine Learning that Matters* (Wagstaff, 2012) asks:

"Many machine learning problems are phrased in terms of an objective function to be optimized. It is time for us to ask a question of larger scope: what is the field's objective function?"

And instead of hyper-focusing on the abstracted proxies to real-world problems such as benchmarks and metrics, she suggests directly reconnecting ML objectives with real-world problems:

"Much effort is often put into chasing after goals in which an ML system outperforms a human at the same task. The Impact Challenges in this paper also differ from that sort of goal in that human-level performance is not the gold standard. What matters is achieving performance sufficient to make an impact on the world."

In the editorial *Machine learning for science and society* (Rudin & Wagstaff, 2013), Cynthia Rudin and Kiri L. Wagstaff ask, "what is Machine Learning good for?" And they point out,

"As things currently stand, it is clear that *our research efforts are not distributed according to the needs of society*. In the communities of machine learning, data mining, and statistics, we spend most of our effort on novel algorithms, novel models, and novel theory, and relatively little effort on the other aspects of the knowledge discovery process, such as data understanding, data processing, feature development, development of new machine learning problems and formulations, practical evaluation and deployment, and how all of these pieces work together to uncover new knowledge in new domains."

Gary Marcus also describes the intensified monoculture in the current AI community, in Chapter 17 *Research Into Genuinely Trustworthy AI* of his book *Taming Silicon Valley: How We Can Ensure That AI Works for Us* (Marcus, 2024):

> 'Ever since 2017 or so, when deep learning began to squeeze out all other approaches to AI, I haven't been able to get the story about the lost keys out of my mind. Often known as the "streetlight effect," the idea is that people typically tend to search where it is easiest to look, which for AI right now is Generative AI.
>
> It wasn't always that way. AI was once a vibrant field with many competing approaches, but the success of deep learning in the 2010s drove out competitors, prematurely, in my view. Things have gotten even worse, that is, more intellectually narrow, in the 2020s, with almost everything except Generative AI shoved aside. Probably something like 80 or 90 percent of the intellectual energy and money of late has gone into large language models. The problem with that kind of intellectual monoculture, is, as University of Washington professor Emily Bender once said, that it sucks the oxygen from the room. If someone, say, a graduate student, has a good idea that is off the popular path, probably nobody's going to listen, and probably nobody is going to give them enough money to develop their idea to the point where it can compete. The one dominant idea of large language models has succeeded beyond almost anyone's expectations, but it still suffers from many flaws, as we have seen throughout this book.'

This monoculture tends to exclude diverse works that pose different visions of AI and deviate from the mainstream. For example, Rudin & Wagstaff (2013) mention the issue of monoculture in the publication and career advancement system:

> "For a researcher to work on applied problems with no top avenue for publication leads to problems with career advancement. This could (and already does) present researchers with an unfortunate choice: work on problems that are either important to society or beneficial to their own career, but not both."

The following excerpt from Jenna Burrell and Jacob Metcalf's editorial *Introduction for the special issue of "Ideologies of AI and the consolidation of power": Naming power* (Burrell & Metcalf, 2024) also reflects the phenomenon that works with different imaginations about the purposes and outcomes of AI could be not welcomed and rejected by mainstream venues in the AI community:

> "This special issue arises from these concerns about the consolidation of power in AI, and the direct experience that the editors and authors of the articles in this collection have with the challenges of naming this power inside computer science research venues. Indeed, as we explain below, a common thread across these papers is that they both name mechanisms by which AI consolidates power and most were rejected at least once in research venues that publish interdisciplinary and/or mainstream computer science research. Such research fits awkwardly within the norms, conventions, and expectations of what and who computer science research is for. While the reasons any paper is rejected for publication is multiply-determined, these rejections raise questions about whether critical research about the purposes and outcomes of AI can be fairly reviewed and, ultimately, find a place in the venues that publish AI research. We, the guest editors, consider the publication of critique to be important for the integrity of the research field."

To see why the high-level critical reflection on the objectives and imaginations of AI matters, we would like to refer to a case in obesity science. In her book *Soda Science: Making the World Safe for Coca-Cola* (Greenhalgh, 2024), anthropologist Susan Greenhalgh tells the story of how industry leader Coca-Cola collaborated with academia to conduct real scientific research that advocated exercise, not calorie restraint, as the priority solution for obesity. This distorted research agenda influenced public health policies on obesity and public understanding on diet and lifestyle in favor of the needs and profits of soda industry. In this case, by exposing and criticizing soda industry's tactics, it is not meant to deny the value of conducting scientific research on exercise, but to show that setting exercise as the mainstream priority is a tactic to downplay other more important factors that the powerful players don't encourage us to focus on, which is calorie intake. This is not a solo case. Other studies in the pharmaceutical industry also show how private-oriented power can distort the research agenda, influence the integrity and independence of research, and shape health policies, public understanding on diseases, and medical practice (Sismondo, 2018; Michaels, 2020). The following quotes from the book *Ghost-Managed Medicine: Big Pharma's Invisible Hands* (Sismondo, 2018) outline how ordinary research activities can be shaped to server for the private, not public, interests. There are also other notorious examples from the tobacco and energy industries (Brandt, 2009; Oreskes & Conway, 2011).

"Pharmaceutical companies sustain large networks to gather, create, control and disseminate information. They provide the pathways that carry this information, and the energy that makes it move. Through bottlenecks and around curves, knowledge is created, given shape by the channels it navigates. Pharma companies create medical knowledge and move it to where it is most useful; much of it is perfectly ordinary knowledge that happens to support their marketing goals. But because of the companies' resources, their interests and their levels of control, they become key shapers of almost all medical terrains."[9]

"Together, the many elements that pharmaceutical companies shape, adjust and assemble constitute markets. These markets are new creations, but because they draw together medical science and health needs they take on an appearance of necessity. They look like entities that have emerged whole from just below the social surface.

The goal of pharma's assemblage marketing is to establish conditions that make specific diagnoses, prescriptions and purchases as obvious and frequent as possible. Ideally, all of the elements of a market can be directed towards the same issues, claims and facts, so that the drugs sell themselves. Pharma companies can then recede into the background, and apply only minimal pressure when needed."[10]

Given the close relationship between AI academia and industry today (Ahmed et al., 2023; Ochigame, 2022; Young et al., 2022; Birhane et al., 2022) and numerous evidence of AI harms (Crawford, 2021; Gray & Suri, 2019; Resnikoff, 2021; Rosenblat, 2018; O'Neil, 2016; Noble, 2018; Costanza-Chock, 2023; Atanasoski & Vora, 2019b; D'Ignazio & Klein, 2020; Chun, 2021; Schaake, 2024; Zuboff, 2019; Eubanks, 2018; Broussard, 2018), these cautionary tales provide significant implications for the AI community: How can we ensure that the current mainstream research agenda in AI is not a remake of the above mistakes in healthcare? While the mainstream research agenda predominantly focuses on productivity, performance, and algorithmic novelty, how can we be certain that we are not trapped by suboptimal priority (Brynjolfsson, 2022), which is analogous to the trap in obesity science to prioritize exercise? How can we be certain that there are not any other more important objectives of AI that the powerful directs us away to exclude from the research agenda, akin to the soda industry's downplaying of calorie intake? Can we question the validity and ethics of the mainstream research agenda of AI, to reduce the likelihood of being misled by distorted research agenda and being manipulated for unjust purposes?

To better answer these questions, we can refer to knowledge in technology studies to systematically understand the phenomena around us.

## 2. How does monoculture in the AI community form? Theories and histories from science, technology, and social studies

### The concept of sociotechnical system

Although people may tend to assume that technological innovations happen in a free space that allow people to explore different visions and directions at their will, the above phenomenon of monoculture in AI shows that the reality could hardly be the case. This is because technological research and development are not merely personal endeavors carried out by "lone genius inventor" (Matthewman, 2011), they are also social processes and are influenced by complex social factors, such as educational and training systems (Rolnick et al., 2024), funding agencies (Young et al., 2022), career incentives (Rolnick et al., 2024; Rudin & Wagstaff, 2013), and peer review and publication systems (Rolnick et al., 2024; Rudin & Wagstaff, 2013; Andrews et al., 2024; Lipton & Steinhardt, 2019), as pointed out by many previous works in the AI community. These social activities are also embedded in a larger social context where technological innovation cannot be isolated from. Science, Technology, and Society (STS) studies show that "technologies can not be abstracted from the environments which they help to create" (Matthewman, 2011). In this sense, arguing whether technology is value-free, value-neutral, or apolitical is meaningless, because the full lifecycle of technology — its creation, deployment, and use — are always embedded in and inseparable from their social contexts. Therefore, we need to consider the specific social, political, and historical dimensions when understanding and analyzing technology. This is the concept of sociotechnical system in STS that "concerns the ways in which people and technologies are irrevocably entangled across different scales such that human agency is intertwined with technological devices, practices, and infrastructures. ...The lens of sociotechnical systems adds the crucial dimension of thinking about how human and technical capacities to act are intertwined. ...[It] invites practitioners to interrogate the sociotechnical systems in which one's project intervenes, which may involve considering the systems upon which the project depends in order to function and also how the output of one's analysis can instantiate novel sociotechnical arrangements" (Boenig-Liptsin et al., 2022). Failure to consider the social dimension can "render technical interventions

---

[9]From Sismondo (2018), Chapter 1 Power and Knowledge in Drug Marketing
[10]From Sismondo (2018), Chapter 8 Conclusion: The Haunted Pharmakon

ineffective, inaccurate, and sometimes dangerously misguided" (Selbst et al., 2019).

## Technology and society as co-production

The way technology and society intertwine with each other naturally leads to the second concept that technology and society co-evolve and co-produce each other. The co-production concept requires us to think beyond the technological or social deterministic mindset (Matthewman, 2011). It helps us to understand the source of harms, as well as providing ways for responsible actions in technology and society. As Margarita Boenig-Liptsin, Anissa Tanweer, and Ari Edmundson put it in their paper *Data Science Ethos Lifecycle: Interplay of Ethical Thinking and Data Science Practice* (Boenig-Liptsin et al., 2022):

> "[I]t encourages practitioners to consider how their questions, processes and outputs are shaped by the social milieus in which they are produced *and* how they simultaneously shape the social world by, for example, occasioning new routines and practices..., affording certain behaviors while constraining others..., or distributing agency and accomplishment... . Thinking in terms of sociotechnical systems probes how technological artifacts reflect and reproduce the social contexts of their production, how human flourishing can be considered in the design of technology, and how accountability can be achieved when risks and responsibilities are distributed widely and opaquely across human and nonhuman actors."

Ruha Benjamin also explains the concept of co-production and its relationship with imagination, in her book chapter *Introduction: Discriminatory Design, Liberating Imagination* (Benjamin, 2019):

> 'In rethinking the relationship between technology and society, a more expansive conceptual tool kit is necessary, one that bridges science and technology studies (STS) and critical race studies, two fields not often put in direct conversation. This hybrid approach *illuminates* not only *how society is impacted by* technological development, as techno-determinists would argue, but how social norms, policies, and institutional frameworks shape a context that make some technologies appear inevitable and others impossible. This process of mutual constitution wherein technoscience and society shape one another is called *coproduction*.
>
> In her book *Dark Matters*, for example, sociologist Simone Browne examines how surveillance technologies coproduce notions of blackness, explaining that "surveillance is nothing new to black folks"; from slave ships and slave patrols to airport security checkpoints and stop-and-frisk policing practices, she points to the "facticity of surveillance in black life." Challenging a technologically determinist approach, she argues that instead of "seeing surveillance as something inaugurated by new technologies ... to see it as ongoing is to insist that we factor in how racism and anti-blackness *undergird* and *sustain* the intersecting surveillances of our present order." Antiblack racism, in this context, is not only a by-product, but a *precondition* for the fabrication of such technologies – antiblack imagination put to work.
>
> A coproductionist analysis calls for more than technological or scientific literacy, but a more far-reaching *sociotechnical imaginary* that examines not only how the technical *and* social components of design are intertwined, but also imagines how they might be configured differently. To extricate carceral imaginaries and their attending logics and practices from our institutions, we will also have to free up our own thinking and question many of our starting assumptions, even the idea of "crime" itself.'

## Embedding social perspective in technology: the concepts of social system, structure, and power

Thinking in sociotechnical system and co-production brings the social dimension into the understanding of technology. A prominent idea in sociology is to utilize systemic thinking that regards **social system**, the whole, larger than a collection of individuals, i.e., a social system contains complex interactions and relationships that cannot be simply reduced to its composing individuals. We will return to this idea in later part on complex systems. As sociologist Allan G. Johnson puts it in his book *The Forest and the Trees: Sociology as Life, Practice, and Promise* (Johnson, 2014):

> "If sociology could teach everyone just one thing with the most profound effect on how we understand social life, it would, I believe, be this: *we are always participating in something larger than ourselves, and if we want to understand social life and what happens to people in it, we have to understand what it is that we are participating in and how we are participating in it*. In other words, the key to understanding social life is neither just the forest

nor just the trees but the forest *and* the trees and the consequences that result from their dynamic relationship to each other.

The larger things we participate in are called social systems, which come in all shapes and sizes. In general, the concept of a 'system' refers to any collection of parts or elements that are connected in ways that coalesce into some kind of whole."[11]

Social systems are like a set of rules in a game that set "paths of least resistance" that affect our thinking and behavior (Johnson, 2014). This systemic perspective allows us to jump out of the individualistic model to understand that the players in the game do not equal to the rules of the game themselves:

"What social life comes down to, then, is a dynamic relationship between social systems and the people who participate in them. Note that people *participate* in systems without being *parts* of the systems themselves. ... This distinction is easy to lose sight of, but it is crucial. It's easy to lose sight of because we are so used to thinking solely in terms of individuals. It is crucial because it means that people are not systems and systems are not people, and if we forget that, we are likely to focus on the wrong thing in trying to solve our problems.

... Thinking of systems as just people is why members of privileged groups often take it personally when someone points out that their society is racist or sexist. ... I cannot make my society or the place where I live suddenly nonracist, but I can decide how to live as a white person in relation to the privileged position of 'white person' that I occupy. ... I can decide what to do about the consequences that racism produces, whether to be part of the solution or just part of the problem. I do not feel guilty because my country is racist, because the creation of racism in this country was not my doing. But as a white person who participates in that society, I feel responsible to consider what to do about it. The only way to get past the potential for guilt and see how I can make a difference is to realize that the system is not I, nor am I the system.

......

For its part, a system affects how we think, feel, and behave as participants. It does this not only through the general process of socialization but also by laying out paths of least resistance in social situations. At any given moment, we could do an almost infinite number of things, but we typically do not realize this and see only a narrow range of possibilities. What the range looks like depends on the system we are in.

.......

This is why people might laugh at racist or heterosexist jokes even when they make them feel uncomfortable—in that situation, to not laugh and risk being ostracized by everyone may make them feel even *more* uncomfortable. The easiest—although not necessarily easy—choice is to go along. This does not mean we *have to* go along or that we *will*, only that if we do go along, we'll run into less resistance than if we don't.

... When we can identify how a system is organized, we can see what is likely to result if people follow the paths of least resistance. We know, for example, where the game of Monopoly is going just by reading the rules of the game. We don't have to know anything about the individuals who play it, except the likelihood that most of them will follow the path of least resistance most of the time.

... People are not systems and systems are not people, which means that social life can produce horrible or wonderful consequences without necessarily meaning that the people who participate are horrible or wonderful themselves. Good people participate in systems that produce bad consequences all the time.

......

Getting clear about the relationship between individuals and social systems can dramatically alter how we see potentially painful issues and ourselves in relation to them. This is especially true for people in privileged groups who otherwise resist looking at the nature and consequences of privilege. Their defensive resistance is probably the biggest single barrier to ending privilege and oppression. Most of the time resistance happens because, like everyone else, people in privileged groups are stuck in an individualistic model of the world and cannot see how to acknowledge white privilege as a fact of social life without also feeling personally to blame for it. And the people who are most likely to feel this way are often the ones who are otherwise most open to doing something to make things better. "[12]

---

[11]From Johnson (2014) Chapter 1 The Forest, the Trees, and the One Thing

[12]From Johnson (2014) Chapter 1 The Forest, the Trees, and the One Thing

Another important concept is **social structure**, which is "[t]he patterns of relationships and distributions that characterize the organization of a social system. Relationships connect various elements of a system (such as social statuses) to one another and to the system itself. Distributions include valued resources and rewards, such as power and income, and the distribution of people among social statuses. Structure can also refer to relationships and distributions among systems" (Johnson, 2014). An important asymmetric relationship in the social structure lies in the concept of **power**, which is defined as "the asymmetric capacity to structure or alter the behavior of others" (Boenig-Liptsin et al., 2022).

Sociological perspective often foregrounds obvious things and allows us to inspect the frame that we are so used to, like the air we breathe. In order to imagine something different, we first have to realize and understand the existence of the frame that bounds our imaginations:

> "Having a set of cultural beliefs allows us to live with a taken-for-granted sense of how things are and to treat the 'facts' of our existence as obvious. What we call 'obvious,' however, is not necessarily what is true. It is only assumed to be true beyond doubt in a particular culture. Without a sense of the obvious, social life loses its predictability, and we lose our basis for feeling secure, but the obvious also blinds us to the possibility that what is 'obviously' true may be false.
>
> ......
>
> I can expand my freedom only by liberating myself from the narrow range of choices that my culture—that any culture—offers the people who participate in it. To do this, I need to step outside the cultural framework I am used to so that I can see it *as* a framework, as one possibility among many. Stepping outside is an important part of what sociological practice is about, and such concepts as culture, beliefs, and values are important tools used in the process, for they point to what we are stepping outside *of*."[13]

> "Sociology ... can change how we see and experience reality by revealing assumptions and understandings that underlie everyday life. Many of these are rarely spoken or otherwise made explicit, and yet they operate in powerful ways to shape our perceptions, thoughts, feelings, and behavior. By going beneath the surface, sociology reveals a deeper and more complex reality."[14]

We bring these perspectives into the following understanding of the history of AI culture.

**Historical lens on the formation of monoculture in AI**

Historical studies show that the AI field did not develop in a vacuum, but was closely related to the social, economic, and political power, ideologies, and values at a given time. Scholars have traced the linkage between AI and ideologies such as eugenics, racism, colonialism, anti-democratic and anti-humanist utopianism (Burrell & Metcalf, 2024; Ali et al., 2023). It should be noted that although the majority in the AI community may not be aware of or do not agree with these ideologies, it is the power and historical contingency that enabled a few privileged people, not the public or the majority in the community via a democratic process, to influence the research agenda and the underlying ideologies that set the paths of least resistance in the AI community. As the previous part indicates, once the rules of the game are set, we can know where the game is going without any information about the game players. And the rules of the game can appear to be necessary, obvious, natural, and inevitable. As Jenna Burrell and Jacob Metcalf put it in their editorial *Introduction for the special issue of "Ideologies of AI and the consolidation of power": Naming power* (Burrell & Metcalf, 2024):

> 'When we raise the question of consolidation of power through AI, we are pointing to a narrowing of who is able to shape the conditions of life, and how we understand ourselves in relation to others, through the operation of technoscientific power. As multiple papers in this collection explore, central to the ideology, economy, and research of AI is the presumption that a small and largely homogenous group of people have the legitimate ability to decide what is "good" for "humanity," sometimes on an epochal and stellar scale. Many billions of dollars have been leveraged to create research agendas and products based on the speculative whims of white, North American men (with a concentration in a handful of Californian postal codes as well as a couple of centers at Oxford University) who are arguing about the optimal way to replace humanity with consciousnesses that live on computer chips in distant galaxies because it is what "the universe" intends (Metz, et al., 2023). At times, the hubris is as astonishing as it is bizarre. Of course, most people who do the day-to-day research and product

---

[13]From Johnson (2014) Chapter 2 Culture: Symbols, Ideas, and the Stuff of Life
[14]From Johnson (2014) Chapter 6 Things Are Not What They Seem

work in this space are not oriented toward distant hypothetical human-free futures, but as Gebru and Torres in this issue point out, the owners and founders of the most prominent labs *are*. And as Ahmed, et al. in this issue demonstrate, there is a deliberate network to recruit and support young research talent to take up careers that serve these eugenic fantasies.'

They provided two examples to show how these ideologies are smuggled into the mainstream using seemingly neutral and scientific language:

> Several contributions in this special issue ... show that dispassionate and scientific sounding language is useful for smuggling old and rightly rejected ideas back into the mainstream. This is a critical point made powerfully in Gebru and Torres' account of how eugenicist thinking has made inroads in AI research. They track the conceptual genealogy and social and political organization of the AGI research agenda (aka the 'TESCREAL bundle') through transhumanist and eugenicist — and often frankly racist — intellectual traditions. Over time, language around race in the United States and Europe has changed and certain ways of characterizing and stereotyping racial groups is no longer publicly acceptable. Yet people still find ways to assert a notion of natural human hierarchy hinged on dubious measures of 'intelligence' which serves, for some adherents, as a thin veil over a belief in racial hierarchy. Much like older forms of eugenic thinking, power is exercised by a small group of people deciding what is "best" for "humanity," invoking along the way a preferred notion of nature, God, or universal teleology to justify why they should build a world that excludes certain types of human life. These beliefs find a new language that avoids old triggers and in this way can attract new (perhaps unwitting) adherents. Birhane, et al.'s contribution similarly shows that asserting robots' rights, which gives the appearance of a highly novel concern, has the regressive effect of diluting efforts to defend and protect universal rights that are unique to *humans*. Additionally, behind these robots stand the interests of the large corporations which build them. Thus pursuing robots' rights by establishing a non-human entity on equal footing with humans threatens to undercut the public interest where it stands opposed to corporate bodies.

A number of works have identify the goal of AI and its roots in eugenics. In their book *How Data Happened: A History from the Age of Reason to the Age of Algorithms* (Wiggins & Jones, 2023), Chris Wiggins and Matthew L. Jones trace the historical genesis of the contemporary statistics and data science in figures such as Francis Galton, who is the originator of eugenics, and Karl Pearson, who institutionalized eugenics. In her book *Discriminating Data: Correlation, Neighborhoods, and the New Politics of Recognition* (Chun, 2021), Wendy Hui Kyong Chun reveals the ties between correlation in machine learning and its eugenic history: both "seek to tie the past to the future—correlation to prediction—through supposedly eternal, unchanging biological attributes." In their paper *The TESCREAL bundle: Eugenics and the promise of utopia through artificial general intelligence* (Gebru & Torres, 2024), Timnit Gebru and Émile P. Torres trace the field's current unquestioned motivation for artificial general intelligence (AGI) back to the normative framework of the newer version of eugenics pushed by a few powerful leading figures. The eugenic ideals are dubbed the "TESCREAL bundle," which denotes "transhumanism, Extropianism, singularitarianism, (modern) cosmism, Rationalism, Effective Altruism, and longtermism." These utopia goals justify the creation of unsafe algorithms, concentration of power, evasion of accountability, and ongoing harms to marginalized groups upon which AI is built. Furthermore, such ideologies are naturalized so that they appear as inevitable research agendas in the AI field to attract unwitting researchers and practitioners, as Timnit Gebru and Émile P. Torres describe:

> 'The AGI race is not an inevitable, unstoppable march towards technological progress, grounded in careful scientific and engineering principles (van Rooij, et al., 2023). It is a movement created by adherents of the TESCREAL bundle seeking to "safeguard humanity" (Ord, 2020) by, in Altman's words, building a "magic intelligence in the sky" (Germain, 2023), just like their first-wave eugenicist predecessors who thought they could "perfect" the "human-stock" through selective breeding (see Bloomfield, 1949). Through concerted campaigns to influence AI research and policy practices backed by billions of dollars, TESCREALists have steered the field into prioritizing attempts to build unscoped systems which are inherently unsafe, and have resulted in documented harms to marginalized groups.
>
> ... This investment has succeeded in legitimizing the AGI race such that many students and practitioners who may not be aligned with TESCREAL utopian ideals are working to advance the AGI agenda because it is presented as a natural progression in the field of AI. In the same way that first-wave eugenicists and race scientists sought and

achieved academic legitimacy for their research (Saini, 2019), TESCREALists have created a veneer of scientific authority that makes their ideas more palatable to uncritical audiences, and thus have succeeded in influencing research and policy directions in the field of AI. First-wave eugenics proved to be ineffective and catastrophic. But as Jean Gayon and Daniel Jacobi signify with the term "eternal return of eugenics," eugenic ideals keep on being repackaged in different forms. The AGI race is yet another attempt, diverting resources and attention away from potentially useful research directions, and causing harm in the process of trying to achieve a techno-utopian ideal crafted by self appointed "vanguards" of humanity.'

Another line of historical lens traces the imperial, capitalist, and colonial roots in AI that questions the "origin myths that treat AI as a natural marriage of logic and computing in the mid-twentieth century" (Ali et al., 2023). Ali et al. (2023) in their editorial *Histories of artificial intelligence: a genealogy of power* illustrate that the purported different periods in AI history have "shared logics – especially managerial, military, industrial and computational – cut across them, often in ways that reinforce oppressive racial and gender hierarchies." In the paper *Animo nullius: on AI's origin story and a data colonial doctrine of discovery* (Penn, 2023), Jonnie Penn "traces elements of the theoretical origins of artificial intelligence to capitalism, not neurophysiology," and compares data colonialism and privatization of the data commons with historical colonialism. The enclosure of the data commons is naturalized and veiled via the scientific pedigree of AI. In his book *Artificial Whiteness: Politics and Ideology in Artificial Intelligence*, Yarden Katz inspects the history of AI and its formation in the American military-industrial complex. The nebulous concept of AI helps its promoters to continuously reinvent and rebrand it to serve the aims of empire and capital (Katz, 2020). In the paper *The Problem with Intelligence: Its Value-Laden History and the Future of AI*, Stephen Cave summarizes the value-laden history of intelligence, and the entanglement of intelligence with the matrices of domination, patriarchy, colonialism, scientific racism, and eugenics (Cave, 2020).

These historical, social, and political perspectives are embedded in the following supplementary references on the critical examination of the mainstream imagination of AI.

## B.2. Section 2. The mainstream imagination of AI (MIA) is not as ethical as we assume

### B.2.1. S1. AI CAN BE OMNISCIENT

**Philosophy and worldviews of science and technology**

In their book *The Blind Spot: Why Science Cannot Ignore Human Experience* (Frank et al., 2024), Adam Frank, Evan Thompson, and Marcelo Gleiser argue that the crises of modern science and technology lie in the underlying problematic philosophy and worldview, which they name the Blind Spot worldview. The Blind Spot worldview is a "culturally ubiquitous mind-set", "like the air we breathe."

> '[F]or most people, including most scientists, [the Blind Spot] it's so pervasive that it doesn't seem like philosophy at all. Rather, people think it's just "what science says."... In truth, however, it's not what science says. Instead, it's an optional metaphysics attached to, but separable from, the actual practice of science. There are other and better ways to understand the relationship between science and the world, as we argue here.'[15]

> 'We call the source of the meaning crisis the Blind Spot. At the heart of science lies something we do not see that makes science possible, just as the blind spot lies at the heart of our visual field and makes seeing possible. In the visual blind spot sits the optic nerve; in the scientific blind spot sits direct experience—that by which anything appears, shows up, or becomes available to us. It is a precondition of observation, investigation, exploration, measurement, and justification. Things appear and become available thanks to our bodies and their feeling and perceiving capacities. Direct experience is bodily experience. "The body is the vehicle of being in the world," says French philosopher Maurice Merleau-Ponty, but as we will see, firsthand bodily experience lies hidden in the Blind Spot.'[16]

> "We have argued that we must inscribe ourselves back into the scientific narrative as its creators. Science rests on how we experience the world. There is no way to take ourselves out of the story and tell it from a God's-eye perspective. Forgetting this fact means succumbing to the Blind Spot, and that means losing our way in both science and all the critical ways that science shapes society."[17]

---

[15]From Frank et al. (2024) Chapter 1 The Surreptitious Substitution: Philosophical Origins of the Blind Spot
[16]From Frank et al. (2024) Chapter An Introduction to the Blind Spot
[17]From Frank et al. (2024) Chapter Afterword

In the later part of the Introduction of the book, Frank et al. (2024) provide a parable of the bodily experience of warm and cold and the scientific concept and measurement of temperature to illustrate the Blind Spot worldview:

> 'The Blind Spot arrives when we think that thermodynamic temperature is more fundamental than the bodily experience of hot and cold. This happens when we get so caught up in the ascending spiral of abstraction and idealization that we lose sight of the concrete, bodily experiences that anchor the abstractions and remain necessary for them to be meaningful. The advance and success of science convinced us to downplay experience and give pride of place to mathematical physics. From the perspective of that scientific worldview, the abstract, mathematically expressed concepts of space, time, and motion in physics are truly fundamental, whereas our concrete bodily experiences are derivative, and indeed are often relegated to the status of an illusion, a phantom of the computations happening in our brains.
>
> ......
>
> The downplaying of our direct experience of the perceptual world while elevating mathematical abstractions as what's truly real is a fundamental mistake. When we focus just on thermodynamic temperature as an objective microphysical quantity and view it as more fundamental than our perceptual world, we fail to see the inescapable richness of experience lying behind and supporting the scientific concept of temperature. Concrete experience always overflows abstract and idealized scientific representations of phenomena. There is always more to experience than scientific descriptions can corral. Even the "objective observers" privileged by the scientific worldview over real human beings are themselves abstractions. The failure to see direct experience as the irreducible wellspring of knowledge is precisely the Blind Spot.'

Specifically, the Blind Spot worldview is an constellation of the following pathologies:

> "**1. Surreptitious substitution.** This is the replacement of concrete, tangible, and observable being with abstract and idealized mathematical constructs. Besides the parable of temperature, other examples we will discuss are substituting clock time for duration, nature at an instant for nature as process, computation for meaning, and information for consciousness. The surreptitious substitution is essentially the replacement of being with the products of a method for gaining a particular kind of knowledge. It's also the replacement of the life-world and nature outside the scientific workshop with what we manufacture inside the workshop.
>
> **2. The fallacy of misplaced concreteness.** This is the error of mistaking the abstract for the concrete. It underlies the surreptitious substitution.
>
> **3. Reification of structural invariants.** Science produces structural invariants through abstraction from experience in the scientific workshop. They include classification schemes, models, general propositions, logical systems, and mathematical laws and models. They comprise highly distilled residues of experience. Reification happens when they are regarded as essentially nonexperiential things or entities that constitute the objective fabric of reality.
>
> **4. The amnesia of experience.** This happens when we become so caught up in surreptitious substitution, the fallacy of misplaced concreteness, and the reification of structural invariants that experience finally drops out of sight completely. It now resides in the Blind Spot we have created through misunderstanding the scientific method."

Frank et al. (2024) then return to the example of temperature to illustrate how these pathologies manifest in this example:

> "Let's return to the parable of temperature for an illustration. If we say that how hot or cold something feels is subjective and apparent, whereas thermodynamic temperature (the average kinetic energy of atomic motion) is objective and real, we're thinking in terms of the bifurcation of nature. We're surreptitiously substituting an abstraction—a mathematical average, a single number taken as representative of a list of many numbers—for something concrete—an object of sense perception. We're committing the fallacy of misplaced concreteness by treating the abstraction as if it were concrete. We are thereby also reifying a structural invariant of experience. As a result, we've lost sight of direct experience as the source and sustenance of science. We've forgotten the entire process by which structural invariants such as thermodynamic temperature are extracted from but remain residues of direct experience in the scientific workshop. We've succumbed to the amnesia of experience. We are fully ensconced in the Blind Spot."

In book Chapter 7 Cognition, Frank et al. (2024) illustrate how the Blind Spot worldview manifests in the AI field, which they name the computational blind spot:

> 'Assuming that the everyday world consists of definite states that need to be computationally represented inside the head is another instance of the Blind Spot. As the frame problem and the cognitive limitations of deep-learning neural networks indicate, the assumption that the everyday world consists of states with determinate boundaries is a mistake. That assumption is actually a case of surreptitious substitution, of substituting computational states for the imprecise and fluid everyday world.
>
> ......
>
> Two interrelated aspects of what we will call the computational blind spot have now come to light. One is surreptitiously substituting computational models for the everyday world; the other is failing to see how we are led to remake the world so that it gears into the limitations of our computational systems. Both prevent us from seeing how our experience of the world relates to our computational constructions.
>
> ......
>
> We've been arguing that the blind spot of the AI view of the mind is the experience and understanding of meaning. Here, the surreptitious substitution takes the form of substituting computation for genuine embodied intelligence, the fallacy of misplaced concreteness takes the form of treating abstract computational models as if they were concretely real, the bifurcation of nature takes the form of thinking that human intuition (or the experience of meaning or relevance realization) is a subjective epiphenomenon of objective brain computations, and the amnesia of experience takes the form of forgetting that human experience and biases lie deeply sedimented in AI models.
>
> But the computational blind spot extends further. AI or machine learning has the image, in both popular culture and science, of being a science of "pure intelligence" and an entirely technical domain transcending nature. This image belies AI's fundamental dependence on physical nature and socially organized, collective human knowledge. As [Kate] Crawford writes, "Artificial intelligence is both embodied and material, made from natural resources, fuel, human labor, infrastructures, logistics, histories, and classifications." AI is fundamentally an "extractive industry," one that "depends on exploiting energy and mineral resources from the planet, cheap labor, and data at scale." On the hardware side, AI requires mining rare earth minerals, such as lithium for batteries, and hence demands oil and coal to power huge mines; on the software side, AI demands huge amounts of carbon-producing energy consumption for running high-performance computer-vision, image-recognition, and language-processing programs. In addition, as we argued earlier and as Crawford notes, "AI systems are not autonomous, rational, or able to discern anything without extensive, computationally intensive training with large datasets or predefined rules and rewards." For these reasons, AI or machine learning in no way transcends nature and in no way constitutes a science and technology of extra-human "pure intelligence." The inability to see these facts clearly and the rhetoric belying them are large-scale effects of the computational blind spot.'

The MIA on AI's omniscient prospect can be illustrated from the following excerpt in the book *Artificial Whiteness: Politics and Ideology in Artificial Intelligence* (Katz, 2020): 'An industry of experts from the academic and policy worlds has emerged to interpret AI's significance. According to the experts, AI could grow the economy and free people from labor burdens, help address climate change, remove the "bias" from the courts of law, automate scientific discovery, reinvent journalism (and hence democracy), and potentially produce, not long from now, "superhuman" intelligent machines that may launch new "civilizations" across the galaxy.'

The MIA manifests scientism or scientific triumphalism, which is criticized in *The Blind Spot* book:

> '[S]cientific triumphalism doubles down on the absolute supremacy of science. It holds that no question or problem is beyond the reach of scientific discourse. It advertises itself as the direct heir of the Enlightenment. But it simplifies and distorts Enlightenment thinkers who were often skeptical about progress and who had subtle and sophisticated views about the limits of science. Triumphalism's conception of science remains narrow and outmoded. It leans heavily on problematic versions of reductionism—the idea that complex phenomena can always be exhaustively explained in terms of simpler phenomena—and crude forms of realism—the idea that science provides a literally true account of how reality is in itself apart from our cognitive interactions with it. Its view of objectivity rests on an often unacknowledged metaphysics of a perfectly knowable, definite reality existing "out there," independent of our minds and actions. It often denies the value of philosophy and holds that

more of the same narrow and outmoded thinking will show us the way forward. As a result, theoretical models become ever more contrived and distant from empirical data, while experimental resources are applied to low-risk research projects that eschew more fundamental questions. Like Victorian-era spiritualism and pining for ghosts, scientific triumphalism looks backward to the fantasized spirit of a long-dead age and cannot hope to provide a path forward through the monumental challenges that science and civilization face in the twenty-first century.'

### Situated knowledge, epistemic injustice, and the theory-free myth of AI

An example of ignoring human experience and denying human's indispensable epistemic footing in AI is the myth of value-free and theory-free AI, which state that AI model and its data can be free from their contexts, human values, knowledge, theories, and interventions to represent the ideal neutral and objective knowledge, a "view from nowhere" (Nagel, 1989). For example, Chin-Yee & Upshur (2019) point out that '[p]roponents of big data and machine learning retort that the "atheoretical" perspective is in fact a strength of these methods, avoiding preconceptions that stifle scientific progress. Some pundits have gone as far as to declare the "end of theory" in the era of big data (Anderson 2008).'

The contemporary development in epistemology, especially feminist epistemology, argues against this outmoded perspective. Discussions are usually under the term of "theory-ladenness of observation" (Chin-Yee & Upshur, 2019). In her influential paper *Situated Knowledges: The Science Question in Feminism and the Privilege of Partial Perspective* (Haraway, 1988), Donna Haraway denies the knowledge of "God trick" "of being nowhere while claiming to see comprehensively," and instead proposes the feminist objectivity of situated knowledge, the knowledge from partial perspective:

'Above all, rational knowledge does not pretend to disengagement: to be from everywhere and so nowhere, to be free from interpretation, from being represented, to by fully self-contained or fully formalizable. Rational knowledge is a process of ongoing critical interpretation among "fields" of interpreters and decoders. Rational knowledge is power-sensitive conversation. ... So science becomes the paradigmatic model, not of closure, but of that which is contestable and contested. Science becomes the myth, not of what escapes human agency and responsibility in a realm above the fray, but, rather, of accountability and responsibility for translations and solidarities linking the cacophonous visions and visionary voices that characterize the knowledges of the subjugated. ... We seek not the knowledges ruled by phallogocentrism (nostalgia for the presence of the one true Word) and disembodied vision. We seek those ruled by partial sight and limited voice—not partiality for its own sake but, rather, for the sake of the connections and unexpected openings situated knowledges make possible. Situated knowledges are about communities, not about isolated individuals. The only way to find a larger vision is to be somewhere in particular. The science question in feminism is about objectivity as positioned rationality. Its images are not the products of escape and transcendence of limits (the view from above) but the joining of partial views and halting voices into a collective subject position that promises a vision of the means of ongoing finite embodiment, of living within limits and contradictions—of views from somewhere.'

Because everyone has a unique perspective and contribution to collective knowledge, downplaying the role of some knowers in favor of other knowers is the manifestation of epistemic injustice (Fricker, 2007; Symons & Alvarado, 2022). For example, stating that AI can replace humans is to downplay the epistemic footing of human experience for AI in favor of the epistemic achievement of the AI models and their creators; Failure to acknowledge the intrinsic epistemic limitation of AI that cannot represent all possible perspectives and knowledges is to exclude the knowledges and experience of knowers (usually the marginalized groups) that are not included in AI modeling in favor of knowers whose knowledges are included in the AI model. The latter case is manifested in her book *Artificial Knowing: Gender and the Thinking Machine* (Adam, 1998), where sociologist Alison Adam argues that the "view from nowhere" in AI actually disguises the white, male, middle-class perspective as the natural, universal, and gold standard. Furthermore, Andrews et al. (2024) argue how this value- and theory-free myth has underpinned pseudoscientific and unethical practice in AI.

### Critical complexity

Complex systems provide another perspective to understand why knowledge is situated. In the book *Critical complexity* (Cilliers, 2016), Paul Cilliers describes characteristics of complex systems that consist of a large number of elements that dynamically interact with each other. The interactions are rich, nonlinear, usually have a short range, and there are loops in the interactions. Complex systems are usually open systems, far from equilibrium, and have a history. Since human knowledge and AI modeling often aim to deal with complex systems, understanding complex systems can help us better understand knowledge, as described in (Cilliers, 2016):

'In order to predict the behaviour of a system accurately, we need a detailed understanding of that system, i.e., a model. Since the nature of a complex system is the result of the relationships distributed all over the system, such a model will have to reflect all these relationships. Since they are nonlinear, no set of interactions can be represented by a set smaller than the set itself – superposition does not hold. This is one way of saying that complexity is not compressible. Moreover, we cannot accurately determine the boundaries of the system, because it is open. In order to model a system precisely, we therefore have to model each and every interaction in the system, each and every interaction with the environment – which is of course also complex – as well as each and every interaction in the history of the system. In short, we will have to model life, the universe and everything. There is no practical way of doing this.'[18]

'If one acknowledges the complexity of a system, it becomes more difficult to talk about "natural" boundaries. Boundaries are still required if we want to talk about complex systems in a meaningful way – they are in fact necessary, as argued above – but there are strategic considerations at stake when drawing them. These considerations may include subjective, or intersubjective components, but this does not mean that they are arbitrary. A complex system has structure and patterns that would render some descriptions more meaningful than others, but the point is that we do not have an a priori decision procedure for determining when we are dealing with something "more meaningful". The contingent and historic nature of complex systems entails that our understanding of the system will have to be continually revised; the frames of our models will have to change. The boundaries of complex systems cannot be identified objectively, finally and completely.

This supports the argument that our knowledge of complex systems cannot be reduced to formal algorithms, but has to incorporate considerations of what the knowledge is for. The criteria used to evaluate the knowledge are not independent things; they co-determine the nature of the knowledge (see Rosen 1996). Knowledge cannot be abstract and complete – we cannot "know" something like that. For us to have knowledge about something, it has to be limited.'[19]

### B.2.2. S2. REPLACING HUMAN WITH AI CAN FREE HUMAN

In addition to a series of labor displacements described in the main text, Acemoglu & Johnson (2024) describe in the early stages of the Industrial Revolution, with the introduction of power-loom factories, women and child labor were used to replace craftsmen, which had resulted in the same consequences of cutting wages and gaining more control over workers.

"As weaving became automated, deskilling accompanied disempowerment of the workers. Machines effectively replaced skilled and experienced adult men with women and children, who had less skill and who were also cheaper and easier to control. This reinforced the significantly declining ability of weavers to have a say in their working conditions or the discipline to which they were subjected, and, consequently, control over daily life passed into the hands of employers (see Hammond & Hammond 1919 for further discussion). Of course, this also meant that they had less say in the determination of their pay."

A number of works have shown how the hidden human labor is disguised as the "magic", "intelligence," or "progress" of machines. Neda Atanasoski and Kalindi Vora have a precise description in their book *Surrogate Humanity: Race, Robots, and the Politics of Technological Futures* (Atanasoski & Vora, 2019b):

'Although, in the language of science and technology studies, these technologies are coproduced with the shifting racialized and gendered essence of "the human" itself, promotional and media accounts of engineering ingenuity erase human–machine interactions such that artificial "intelligence," "smart" objects and infrastructures, and robots appear to act without any human attention. These technologies are quite explicitly termed "enchanted"—that is, within technoliberal modernity, there is a desire to attribute magic to techno-objects. In relation to the desire for enchantment, *Surrogate Humanity* foregrounds how this desire actively obscures technoliberalism's complicity in perpetuating the differential conditions of exploitation under racial capitalism.

In the desire for enchanted technologies that intuit human needs and serve human desires, labor becomes something that is intentionally obfuscated so as to create the effect of machine autonomy (as in the example of the "magic"

---

[18]From Cilliers (2016) Chapter What can we learn from a theory of complexity?
[19]From Cilliers (2016) Chapter Knowledge, limits and boundaries

of robot intelligence and the necessarily hidden human work behind it). Unfree and invisible labor have been the hidden source of support propping up the apparent autonomy of the liberal subject through its history, including indentured and enslaved labor as well as gendered domestic and service labor. The technoliberal desire to resolutely see technology as magical rather than the product of human work relies on the liberal notion of labor as that performed by the recognizable human autonomous subject, and not those obscured labors supporting it. Therefore, the category of labor has been complicit with the technoliberal desire to hide the worker behind the curtain of enchanted technologies, advancing this innovated form of the liberal human subject and its investments in racial unfreedom through the very categories of consciousness, autonomy, and humanity, and attendant categories of the subject of rights, of labor, and of property.'

Harry Law also reveals the concealment of labor in the history of early artificial neural networks built in Bell Labs, in the paper *Bell Labs and the 'neural' network, 1986–1996* (Law, 2023):

> "In his history of Charles Babbage's analytical engine, Simon Schaffer argues that fundamental to the persuasive capacity of the rhetoric of 'intelligent' machines is the obscuring of the role of labour. A similar relationship manifested in New Jersey during this period as the minimization of the processes performed by the US Postal Service contractors enabled LeNet's acts of 'recognition' to be linked with the independent action of the artificial neural network. In this way, the contribution of technical and sub-technical labour was shaded by a reduction in the standardized 'error rate' used to demonstrate the effectiveness of the backpropagation algorithm."

The main text discussion of the two groups of people who are replaced by AI and who work behind AI can fit into the big picture of the "institutionalized societal order" of capitalism (Fraser, 2022). In her book *Cannibal Capitalism: How our System is Devouring Democracy, Care, and the Planet – and What We Can Do About It*, philosopher Nancy Fraser provides a framework to deepen our understanding of the capitalist society. In the framework, she expands the foreground economic features of capitalism to their background non-economic preconditions, including "social reproduction, the earth's ecology, political power, and ongoing infusions of wealth expropriated from racialized peoples" (Fraser, 2022). She explores three perspectives of capitalism: exchange, exploitation, and expropriation. The market exchange perspective, characterized by self-interest transactors and growth-and-efficiency-maximizing logic, only narrowly defines capitalism by ignoring its structural dependence on the above background conditions. The backstory of the front story of exchange is exploitation, where workers' surplus labor time is appropriated by capital. The backstory of exploitation is expropriation that makes exploitation possible and profitable, where capitalists brutally confiscate people's assets, including labor, land, minerals, and energy resources. Exploitation and expropriation create two groups of workers, as Fraser states:

> '[T]he distinction between the two exes [exploitation and expropriation] corresponds to a status hierarchy. On the one hand, exploitable "workers" are accorded the status of rights-bearing individuals and citizens; entitled to state protection, they can freely dispose of their own labor power. On the other hand, expropriable "others" are constituted as unfree, dependent beings; stripped of political protection, they are rendered defenseless and inherently violable. Thus, capitalist society divides the producing classes into two distinct categories of persons: one suitable for "mere" exploitation, the other destined for brute expropriation.

The two groups of workers who have been replaced and who replace others can correspond to the two categories of exploitable workers and expropriable workers. Fraser also points out that the distinction between the two categories is blurred in today's society:

> 'In the present regime, then, we encounter a new entwinement of exploitation and expropriation—and a new logic of political subjectivation. In place of the earlier, sharp divide between dependent expropriable subjects and free exploitable workers, there appears a continuum. At one end lies the growing mass of defenseless expropriable subjects; at the other, the dwindling ranks of protected citizen-workers, subject "only" to exploitation. At the center sits a new figure, formally free, but acutely vulnerable: the *expropriated-and-exploited citizen-worker*. No longer restricted to peripheral populations and racial minorities, this new figure is becoming the norm.'

This phenomenon is also manifested in AI replacement, in which workers who are replaced by AI may have to seek more precarious work and suffer from more exploitation/expropriation.

Regarding how automation impoverishes work, Jason Resnikoff has extended evidence and arguments in his book *Labor's End: How the Promise of Automation Degraded Work* (Resnikoff, 2021).

### B.2.3. S3. AI CAN PROMOTE PRODUCTIVITY AND PROSPERITY

Regarding Daron Acemoglu and Simon Johnson's review of the economic history and technical change from the Middle Age to present, we summarize two examples in their book *Power and Progress: Our Thousand-Year Struggle Over Technology and Prosperity* to show how they associate the changes in workers' wages, welfare, working conditions, and directions of technical development with changes in social and political power (Acemoglu & Johnson, 2023).

The first example is described in Chapter 6 Casualties of Progress. In it Acemoglu & Johnson (2023) describe that in Britain, the productivity gain from the early stages of the Industrial Revolution was not shared with workers with uncontrolled pollution and exacerbated public health, until later when workers struggled for political power such as union legalization and the right to vote.

The second example is described in Chapter 7 The Contested Path and Chapter 8 Digital Damage. The shared prosperity in the postwar period in the Western world was due to strong countervailing power that strengthened labor movement and technical development that created new tasks and opportunities for workers of all skill levels. Both conditions collapsed since the 1980s till today, with the rise of a new vision of neoliberalism that shifted the direction of innovation away from worker-friendly technologies to greater automation.

### B.2.4. S4. HARMS OF AI ARE DUE TO MALICIOUS USE

Regarding the discourse that harms of AI are often due to unintended or malicious use, Nassim Parvin and Anne Pollock analyze the political implications of such a discourse in their paper *Unintended by Design: On the Political Uses of "Unintended Consequences"* (Parvin & Pollock, 2020):

> 'On the surface, the term "unintended consequences" captures a rather straight-forward idea: the consequences of technologies or other interventions that were unforeseen at the time of their conception or design. Nobody realized at the time that such and such would follow from what they were doing.
>
> But even a quick survey of the uses of the term in public and scholarly discourse suggests a subtle but significant shift in usage: those consequences of technology that can indeed be anticipated in advance but that fall outside of the purview of the specializations that conceive or implement products. As such, the concept is emptied of its substance while doing substantial work—especially for tech companies and technology developers, as well as their enthusiasts. It provides a category that is descriptive of the social, environmental, and political impacts of science and technology as ones that lack prior, deliberate action. Phenomena described as unintended consequences are deemed too difficult, too out of scope, too out of reach, or too messy to have been dealt with at any point in time before they created problems for someone else. The descriptive approach works as a defensive and dismissive strategy.
>
> .....
>
> It may be viewed positively in that by qualifying those consequences as "unintended," analysts and critics can expose, discuss, or correct such issues without having to assign blame on those who may be responsible for or to them, or having to get into discussions about responsibility that may be too difficult or politically costly. At the same time, the conflation helps those in positions of power to avoid accountability, since they don't have to own the consequences of their choices. It becomes possible for designers and advocates to claim shared values with critics—such as abhorrence of structural inequalities including sexism, racism, or ableism—without taking responsibility for ensuring that the design of technologies upholds those values (see Shelby 2020). ... "Unintended consequences" operates in a subtle but insidious way to uphold the status quo, especially institutional structures and knowledge regimes that have produced those negative impacts—inclusive of the entrenched privileging of expertise in narrowly technical domains—while avoiding accountability.'

Regarding the vision that "AI can fix its own flaws," we refer to the following quote and critique from Jenna Burrell's paper *Automated decision-making as domination* (Burrell, 2024):

> 'In early 2020, just before the COVID-19 pandemic shut down all large gatherings for the forseeable future,

the Association for the Advancement of Artificial Intelligence hosted its annual event. At a keynote panel, AI researchers Yoshua Bengio, Geoffrey Hinton, and Yann LeCun, as recent winners of the Turing Award for lifetime achievement in computing, were given the stage to speak about their technical breakthroughs. A headline summarized the gist of their talks this way: "Deep learning godfathers Bengio, Hinton, and LeCun say the field can fix its flaws" (Ray, 2020). This sentiment captures the essence of disciplinary conservativism. The ritual of posing questions about the future of a research field to its most senior and successful members is not peculiar to AI research or computer science as a discipline. These are realities of how any field that has garnered resources protects its claims to legitimacy (Fourcade, et al., 2015). Yet, one can also imagine the alternatives. What if these senior scholars were to say, "... the field needs outside regulators to set limits" or "... the field requires independent auditors to cast a fresh eye on its flaws" or "Deep learning godfathers ... cede the floor to promising early career scholars ...." While the ostensible purpose of academic fields is to generate new knowledge, they are at the same time social systems; designed to protect resources and aimed ultimately at social reproduction and self-perpetuation.'[20]

This vision often reflects the techno-solutionism or technological fix mindset that ignores the complexity of the problem space and resort to technology as the only source of solution (Morozov, 2014; Tec, 2024).

---

[20]From Burrell (2024) Section 1. Introduction: "The field can fix its flaws"

