# OpenReview forum: "Position: AI for Just Work: Constructing Diverse Imaginations of AI beyond "Replacing Humans''"
_ICML.cc/2025/Position_Paper_Track — Submitted to ICML 2025 Position Paper Track_

### Official Review · Reviewer_UbQM · 2025-02-21

**Significance:** 2
**Argument Clarity:** 2
**Rating:** 2
**Confidence:** 4

**Questions:**

I find the claim about the "mainstream imagination of AI" (MIA) somewhat strange—perhaps it's just me, but this is the first time I've encountered this term along with the assumptions (S1–S4) presented. I also wonder where these assumptions originate, as I don’t believe that **ANY** of my senior colleagues in the AI research community would make a blanket statement like “AI can be used to replace humans” **without specifying a concrete scenario and its constraints**. If the authors are indeed generalizing such claims without context—rather than, for example, discussing how AI might automate tasks like hyperparameter tuning in specific acoustic engineering problems—then I find the framing of this paper even more questionable.

**Discussion Potential:**

1

**Paper Summary:**

This position paper advocates for a broader imagination of AI beyond the prevailing narrative of “Replacing Humans” when considering the fundamental “why” behind AI development. The authors argue that the current ML community operates as a monoculture, shaped by a homogeneous group that prioritizes traditional modeling values such as performance and efficiency, while often overlooking critical perspectives on ethics and the potential harms of AI deployment. Then the authors rebut that the "mainstream imagination of AI" is not ethical and propose a demonstration of how to create new imaginations.

**Position:**

Yes

**Position In Title:**

Yes

**Related Work:**

2

**Strengths And Weaknesses:**

I acknowledge the authors' good intentions in highlighting the potential risks of AI and advocating for a broader discussion on the motivations behind AI development. The proposed demonstration of constructing a new imagination (Section 4) is somewhat interesting to read. Below, I outline the strengths and weaknesses of the paper, though I find the earlier sections particularly questionable.

- First, I do not see sufficient empirical or statistical evidence supporting the core claim that the ML community is a "monoculture shaped by a small and largely homogeneous group of people," aside from the cited paper, *Ideologies of AI and the Consolidation of Power.* If the authors were to directly engage with academic PIs and industry researchers actively working in “mainstream ML research,” they would find that many are well aware of and value diversity and ethical considerations. This is reflected in the substantial body of technical research on fairness, diversity, safety, and trustworthy AI, regularly published in top-tier AI conferences, including ICML.

- Additionally, major conferences like CVPR release detailed statistics on submissions and acceptance rates across different research areas. A quick look at these reports would reveal that tracks related to "responsibility," "trustworthiness," "safety," and "fairness" have become major, independent subject areas. Furthermore, acceptance rates in these tracks are often higher than those in more traditional, modeling-focused tracks such as generative models and representation learning.

- Regarding the authors’ critique of what they refer to as the "mainstream imagination of AI," I find the argument to be unfair and framed in a way that **deliberately assumes the worst interpretation**. For example, summarizing S1 as "AI can be omniscient" seems like a complete misinterpretation of the original statement in Section 2.1 (Also, model self-correction is also an active research field, so I don't it is an false statement in original S1).

- Section 4 presents a more reasonable and engaging perspective, but I am confused by the example of medical image synthesis. To my knowledge, researchers are not actually focused on synthesizing medical images but rather on AI-assisted diagnosis, which is an entirely different problem from generative synthesis.

To be frank, this paper reads more like a critique written **for the sake of criticism**, seemingly from outside the ML research community. It reinforces many stereotypes about ML researchers without engaging in meaningful discussions or investigations into what motivates them and what they are actually working on. Given this, I do not support the acceptance of this paper.

**Support:**

1

---

> ### Author Rebuttal · Authors · 2025-03-31
>
> Dear Reviewer UbQM,
>
> Thank you very much for your time, review, and helpful comments. We appreciate your assessment on the strengths of our paper “in highlighting the potential risks of AI and advocating for a broader discussion on the motivations behind AI development” and the “reasonable and engaging perspective” in Section 4. We also deeply appreciate your concerns and believe that clarifying how we intend our arguments to be received by individual researchers in the community will greatly strengthen this position. Please find our response below.
>
> # 1 Empirical evidence for AI monoculture
>
> We will further clarify two parts of arguments: (1) there is empirical evidence regarding the “values” that are dominant in ML research and (2) we aim to describe values held by particularly powerful actors (e.g. AI labs pursuing “AGI”) and *not* individual researchers.
>
> We provide three empirical evidence in pg 1 line 42-69 right
> * Birhane et al. 2022 analyzes 100 highly-cited papers in ICML and NeurIPS and identifies dominant values
> * Grace et al. 2024 surveys 2778 AI researchers who published in 6 top AI venues on AI progress, and shows “the chance of all human occupations becoming fully automatable was forecast to reach 10% by 2037”
> * Ivanov et al. 2020 shows the phenomenon of societal automation anxiety
>
> We agree with you that individual researchers in our community hold much more diverse and ethical values, but the empirical evidence suggests these values are “outweighed” by more dominant values as described in our paper. Further, while there is great progress in venues like FAccT and AIES, these have less overall influence than “main” ML conferences. We further discuss social structure’s role in Appendix B pg 23-27.
>
> We will clarify this and make minor revisions to (1) avoid the impression that we are making assumptions about the values of any individual ML researchers from outside the ML community, (2) ensure we are highlighting progress in venues like FAccT and (3) make it clear how our position can be beneficial to individual ML researchers who are having values imposed on them by powerful actors.
>
> Our position aims to support researchers by calling the community to change the social structures embedded as collective imaginations. This will enable researchers’ ethical and diverse visions/works to gain more support, recognition, and become a central component of our community.
>
> # 2 Works on responsibility, trustworthiness, safety, and fairness
> We agree with you and acknowledge the ever-increasing works on AI ethics. This doesn't contradict with our motivation to reduce the chances of AI ethics “being trapped by suboptimal objectives” (line 12) to make the community more ethical. Furthermore, the increasing work and good intentions may not prevent ethics washing mentioned in pg 9 line 450, because they may not address the roots of unethical issues. In contrast, we identify the root causes in line 216-227.
>
> # 3 Deliberately assumes the worst interpretation
> Based on your feedback, we will delete “omniscient” and use the five-line S1 statement to replace the short version, which was used due to page limit.
>
> # 4 Task selection in the case study
> Medical AI consists of many different tasks. Although AI-assisted diagnosis may be the most well-known task, it is equally valid to use other tasks such as medical image synthesis. AI-assisted diagnosis is one of the downstream tasks of medical image synthesis. More information on this task can be found in the cited Frangi et al., 2018 paper.
>
> # 5 The “Questions” field in the reviewer form
> The "mainstream imagination of AI" (MIA) is a term we introduce here. The concept of imagination is from social science (line 20). Its background info is in Appendix B pg 18-19. “Mainstream” describes the central tendency of distribution of various collective imaginations as illustrated in Fig.1. More evidence of the dominant imagination is in Appendix B pg 19-21.
>
> Due to page limit, we provide support for S1-S4 in Appendix B.2 or in-text citations.
> * S1: pg 29 line 1577-82
> * S2: Atanasoski & Vora 2019b. Grace et al. 2024. Ivanov et al. 2020
> * S3: Acemoglu & Johnson 2023. Birhane et al. 2022
> * S4: pg 33 line 1810-27
>
> In pg 2 footnote, we acknowledge that actual assumptions in social structures of the community are complex and cannot be fully represented in a few sentences. But in order to discuss them, we have to abstract them. This aligns with the idea of A1 that abstraction is useful but has limitations. We will refine the footnote to acknowledge the limitation more clearly. We also double checked that descriptions of S1-S4 align with the above citations to ensure they’re the best possible abstractions.

---

> > ### Comment · Reviewer_UbQM · 2025-04-04
> >
> > (Accidentally posted this as Official Comment before)
> >
> > I thank the authors for their rebuttal. I have read the responses as well as the comments from fellow reviewers, and some of my concerns have been addressed, such as the statement regarding the monoculture, and MIA. While I understand the authors' intention of "not targeting individual researchers" and the position track papers do not share the same evaluation criteria as the ICML main track, I still have some concerns.
> >
> > First, I think many of the terminologies used, as well as the way the paper is structured and the usage of language, are not very familiar to conventional ICML readers. While this may not necessarily be a bad thing and maybe they are well-established in social science, it does limit the potential discussion and clear understanding of this particular community, also noted by fellow reviewers.
> >
> > Second, while I understand we should not be evaluating position papers with standards like methodology novelty and performance, I still believe a good paper, either in ML or social science, should use rigorous and precise language and statements that are well supported by evidence. While the authors clarify some of them in the rebuttal, I think the overall manuscript is still somewhat concerning.
> >
> > Therefore, I will raise my score to 2 but maybe not higher.

---

> > > ### Author Response · Authors · 2025-04-09
> > >
> > > Dear Reviewer UbQM,
> > >
> > > Thank you very much for your reply; this feedback is deeply helpful for advancing this work. We wanted to clarify the specific revisions we plan to make based on this feedback.
> > >
> > > 1.Based on your feedback on the interdisciplinary nature of the terminologies and language, to support readers’ understanding and discussion on the interdisciplinary topics, we will:
> > >
> > > 1.1 **Highlight the role of Appendix B. Bibliographic Essay on the Multidisciplinary Knowledge Background** in the paper.
> > >
> > > 1.2 **Add a Glossary Table** in the Appendix to facilitate readers’ understanding of the interdisciplinary terminologies in the paper.
> > >
> > > We understand your concern and foresee that the interdisciplinary nature of our arguments may not be the most familiar language to the target readers in the AI community. Therefore, we provided Appendix B that introduced the relevant background knowledge to facilitate readers’ understanding and discussion. In the previous version, we didn’t emphasize the role of Appendix B in the main text of the paper due to page limit, as we only mentioned Appendix B in the footnote in Page 1. Therefore, in the revision, we will add one sentence at the end of Section 2.1, to highlight Appendix B and the newly-added Glossary Table, and emphasize their roles in facilitating readers’ understanding on the interdisciplinary topics discussed in the paper.
> > >
> > > In addition, to help readers better understand the interdisciplinary terminologies, in the revision, we will add a Glossary Table in the Appendix that summarizes the terminologies introduced in the paper. For each piece of terminology, the table will list its definition, its location in the paper, a relevant technical example to instantiate the terminology, and its references.
> > >
> > > We hope these revisions will facilitate AI audiences to better understand and discuss the objectives of AI development regarding their social, ethical, and philosophical roots. We also appreciate your kind consideration that such interdisciplinary discussions in our paper “may not necessarily be a bad thing.” If our hypothesis presented in the paper is correct that the roots of unethical issues in AI originated from factors outside AI techniques in the social and philosophical factors, then such interdisciplinary discussions may not be optional, but essential, for our community to understand such root causes and to appreciate the influence they can have on AI. Such interdisciplinary approaches are also seen in [recent actions to integrate ethics across computer science education](https://cacm.acm.org/research/embedded-ethics/), to enable the technical community to be more familiar with the ethical, social, and philosophical knowledge and languages.
> > >
> > > 2.Based on your feedback on the rigorous and precise use of language and statements, we will **add limitation statements** in the Conclusion. The limitation will acknowledge that, due to paper length, Section 2 offers simplified introductory discussions on the rich and complex disciplines of the philosophy of science, ethics, labor studies, and economic objectives. Each discipline has various and usually competing theories and standpoints on the given topics, and we prioritize perspectives that represent the vulnerable, marginalized, and disproportionately impacted groups over the majority. Thus our arguments are limited by the particular ethical and philosophical standpoint we choose. We hope this limitation declaration can enable readers to better understand our particular perspective and open rooms for discussions from other perspectives.
> > >
> > > Although our arguments in Section 2 only provide a snapshot of the rich discussions in the above disciplines, we try to faithfully summarize the cited references and our reasoning is grounded in the logic of these cited references. This can be shown that in the main arguments of Section 2, almost every one or two sentences are backed up by references, and the succinct arguments in the main text align with the more detailed references in Appendix B. The key justifications in Section 3 and 4 are also backed up by references. Although we read the main evidence of the referred books and papers at least twice for precise summary in Section 2, some omissions may occur due to our limited perspective. We are open to revising specific wording if pointed out by reviewers or readers.
> > >
> > > With the new limitation statements, we hope our paper, although may introduce controversial topics, can serve as a start for the community to openly and critically discuss the objectives and visions in AI development (i.e, the “why” question in AI).

---

### Official Review · Reviewer_K3DY · 2025-03-06

**Significance:** 1
**Argument Clarity:** 4
**Rating:** 1
**Confidence:** 5

**Questions:**

N/A

**Discussion Potential:**

1

**Paper Summary:**

The paper provides a detailed discussion on constructing diverse imaginations of AI. Specificallt, it focuses on the critcal examination such as “AI can be omniscient”, “Replacing human with AI can free human”, and “AI can promote productivity and prosperity”. It also explores serveral AI properties.

# Update After Rebuttal

Thanks for the authors' reply. However, most of the concerns are not solved. Therefore, this paper should be rejected.

**Position:**

No

**Position In Title:**

Yes

**Related Work:**

4

**Strengths And Weaknesses:**

Pros:

1. The paper is very well-written, and it provides an in-depth discussion at the intersection between AI and ethnics.

2. The supplementary is very comprehensive.


Cons:
I only have one concern at this stage. I would like to check the reviews of my fellow reviewers.

The concern is:

Admittedly this is a good paper at the intersection between AI and ethnics, it is not suitable for ICML community, even for the position track. This article appears to be a humanities reseacher’s discussion and analysis of AI. It lacks all of the contributions that we commonly consider, for instance, there are no new method, no new data, no experiments, and no theory. I must admit this paper is good but it is not acceptable in ICML or its position track.

**Support:**

1

---

> ### Author Rebuttal · Authors · 2025-03-31
>
> Dear Reviewer K3DY,
>
> Thank you very much for your time, review, and helpful comments. We appreciate your assessment of our work as being “very well-written, and it provides an in-depth discussion at the intersection between AI and ethics.” We would like to address your concern on the suitability of the paper for the ICML position paper track.
>
> Specifically, regarding your concerns that
> > It lacks all of the contributions that we commonly consider, for instance, there are no new method, no new data, no experiments, and no theory.
>
> According to [the definition from the book *Advanced Writing in English: A Guide for Dutch Authors*](https://books.google.ca/books?id=-gBH7BQqy_0C&pg=PA11&redir_esc=y#v=snippet&q=%22position%20paper%22&f=false), a position paper is a type of paper in which “the writer presents his opinion of events, facts, and experiences and defends that interpretation with arguments.” From your description of our paper as being composed of “discussion and analysis of AI” and “the critical examination”, it seems our paper fulfills the definition of a position paper.
>
> In contrast, the “new method, new data, experiments, and theory” you mentioned are the features of an empirical or theoretical research paper. The [ICML position paper track instruction page](https://icml.cc/Conferences/2025/CallForPositionPapers) highlights the difference between empirical research paper and position paper as follows:
> > The review criteria for position papers differs from those of the main conference track.  Submissions to the main ICML conference track emphasize original research and novel results.  In contrast, submissions to the position paper track will be judged primarily on whether they present a compelling position that warrants greater exposure within the machine learning community
>
> Regarding our specific position we call for in this paper, we have clarified it in our response to the second reviewer, Reviewer CaDp, in **Part # 1. Paper’s central position** for your reference.
>
> We also note that in our response to the first reviewer, Reviewer VH64, in **Part # 1. Regarding “include more concrete technical methodologies”**, we clarify our concrete examples of how taking the position seriously here would materially change the practices and outcomes of ML researchers and practitioners in contexts like health care.

---

### Official Review · Reviewer_CaDp · 2025-03-17

**Significance:** 2
**Argument Clarity:** 2
**Rating:** 2
**Confidence:** 3

**Questions:**

What is the paper’s central position?

What are the key evaluation metrics for "Constructing Diverse Imaginations of AI"? How should researchers measure success?

**Discussion Potential:**

2

**Paper Summary:**

The paper explores the “why” question of AI and diversifies the imagination of AI beyond replacing humans. To illustrate their approach of constructing a new imagination of “AI for just work”, they apply it to medical image synthesis, demonstrating how AI can be designed with ethical and worker-centered principles.

**Position:**

No

**Position In Title:**

No

**Related Work:**

2

**Strengths And Weaknesses:**

Strengths:
The paper is well-supported with references.
The discussion is likely to inspire debate about AI’s role in labor, fairness, and ethical decision-making.

Weaknesses:
The paper’s position is unclear—while it critiques the dominant AI paradigm and suggests an alternative, it presents more like a proposal of a new method rather than in favor or against a particular research priority, a call to action, a value statement, a policy proposal, or a recommendation for changes. It would be helpful for the authors to present their position more clearly rather than emphasizing the methodology of Constructing Diverse Imaginations of AI.

**Support:**

2

---

> ### Author Rebuttal · Authors · 2025-03-31
>
> Dear Reviewer CaDp,
>
> Thank you very much for your time, review, and helpful comments. We appreciate your assessment of our work that “The discussion is likely to inspire debate about AI’s role in labor, fairness, and ethical decision-making,” which affirms that our contribution aligns with the goal of ICML position paper track. Please find our responses to your questions below.
>
> # 1. Paper’s central position
>
> ## The central position of the paper is stated in the following positions:
>
> * Abstract
> > We then call to diversify our collective imaginations of AI, embedding ethical assumptions from the outset in the imaginations of AI.
>
> * Introduction page 2, line 86-87, in bold
> > We call for 1) diversified collective imaginations of AI that 2) synergize both ethical values and AI development.
>
> * TL;DR in the submission form (Executive summary)
> > The current AI community is dominated by a homogeneous vision of AI, which underlying assumptions may not be as ethical as we assume. We call for diversified collective imaginations of AI that synergize both ethical values and AI development.
>
> ## Our call to actions
> The paper aims to call on the AI community to deviate from “the dominant AI paradigm” (quoting your comment) that prioritizes research objectives such as replacing humans and improving productivity, and call for **diverse AI paradigms** that prioritize research agendas that hold various visions/imagination of AI and embed ethical values from the outset in the imagination of AI. Our call for actions or recommendations for change in the AI community is stated in the last sentence in *Section 3 Ethical and diverse imaginations are needed for AI* to create research cultures and agendas that provide more support for research projects with these diverse imaginations of AI.
>
> ## Why Section 4 is necessary
> Sections 1-3 is the structure of a typical position paper, with an alternative view and our counterarguments to it in Section 2, and our call for a new position on “what should be done” in Section 3.  Section 4 is on “how to get it done.” We emphasize the problem of “how to get it done” because our paper deals with the special topic of imagination that naturally resists challenge and change. Imagination provides the deep and unconscious frame of thinking for us to construct a taken-for-granted reality, like the air we breathe (more details on the concept of imagination from social science are introduced in Appendix B.1.1, page 18-19). Because of this, being able to constantly realize the frame of our thinking, have conscious reflection of our deep background assumptions, and be able to change them are a very difficult endeavor, as we have described in page 5 line 220-223 right:
>
>
> > ... individuals and the community need to constantly investigate efforts in collective imaginations to “get over some of this deeply habituated laziness and start engaging in interpretive (imaginative) labor for a very long time to make those realities stick”
>
> Therefore, by providing a methodological demonstration of the process of taking actions and showcase what a new imagination and its implementation look like, Section 4 provides more support to make our call easier to be converted to real actions, rather than staying as a looking-good-but-hard-to-implement call.
>
> # 2. What are the key evaluation metrics for "Constructing Diverse Imaginations of AI"? How should researchers measure success?
>
> To evaluate how ethical and diverse the imaginations are in the AI community, we need two steps of method (M1,M2) to:
>
> * M1. Identify and assess the fulfillment on ethics
> * M2. Assess diversity
>
> For M1, we can examine whether key features of AI ethics appear in a project, such as:
> * consideration and protection of the disproportionately impacted groups
> * embedding of values and benefits from the less powerful stakeholders
> * necessary precautions that set and assess the scope and limitations of the proposed technique to avoid harms
>
> We can use paper annotation methodology to identify the ethical features for a paper. The following two works provide methodologies on annotating papers based on AI ethical features or values encoded in AI projects:
>
> * [The Forgotten Margins of AI Ethics](https://dl.acm.org/doi/10.1145/3531146.3533157)
> * The cited Birhane et al., 2022 paper
>
> For M2, from the identified work in M1, we can use some quantitative metrics to measure the degree of diversity regarding various directions on the ethical objectives/imaginations.
>
> Based on your comment, in our revision, we will extend the above methodological procedure as a preliminary evaluation framework in the Appendix. We will mention this framework in Conclusion, and point out future research directions to explore different assessment methodologies and to conduct more empirical works to assess the progress of our call.

---

### Official Review · Reviewer_VH64 · 2025-03-18

**Significance:** 3
**Argument Clarity:** 3
**Rating:** 3
**Confidence:** 3

**Questions:**

See weakness.

**Discussion Potential:**

3

**Paper Summary:**

This paper critically examines the dominant "mainstream imagination of AI" (MIA), which emphasizes replacing human labor, improving productivity, and achieving technical progress, often at the expense of ethical considerations. The authors argue that this approach perpetuates inequality, devalues labor, and creates unjust hierarchies. They advocate for the diversification of collective imaginations of AI and propose a new framework, "AI for just work," which prioritizes workers' rights, ethical values, and real-world grounding. The paper demonstrates how this new imagination can guide technical decisions, using medical image synthesis (MISyn) as a case study. Key recommendations include embedding ethical assumptions, conducting thorough limitation analyses, and empowering workers in AI development processes.

**Position:**

Yes

**Position In Title:**

Yes

**Related Work:**

3

**Strengths And Weaknesses:**

This paper highlights the ethical and societal shortcomings of AI while introducing a new framework. If the paper could include more concrete technical methodologies and real-world examples, its impact would be even more significant. Overall, this paper is likely to inspire meaningful discussions within the community.

**Support:**

3

---

> ### Author Rebuttal · Authors · 2025-03-31
>
> Dear Reviewer VH64,
>
> Thank you very much for your time, review, and helpful comments. We  appreciate your assessment of our work that “*this paper is likely to inspire meaningful discussions within the community*,” which acknowledges that our contribution aligns with [the goal of ICML position paper track](https://icml.cc/Conferences/2025/CallForPositionPapers) “to highlight papers that stimulate (productive, civil) discussion on timely topics that need our community’s input.”
>
> We would like to answer your question on:
> > “If the paper could include more concrete technical methodologies and real-world examples, its impact would be even more significant.”
>
> # 1. Regarding “include more concrete technical methodologies”
>
> Due to the page limit, the concrete changes of technical practice before and after the new imagination is provided in *Appendix A Table 1* for the case of the medical image synthesis task in Section 4.3. In the table, we describe the detailed changes of practice on technical and non-technical aspects.
>
> The changes in practice of technical methodologies listed in App. A -Table 1 include:
>
> 4. Evaluation reporting standard
> 5. Evaluation scope
> 6. Limitation analysis
>
> The changes of non-technical practice listed in App. A - Table 1 include:
> 1. Risk prevention of misinformation
> 2. Power to reject AI
> 3. Problem formulation
>
> # 2. Regarding “include more real-world examples”
>
> Since we call for diverse and ethical imaginations of AI, in addition to the existing example in Section 4.3, to reflect the diversity of the new imaginations, we have provided an additional example of the Indigenous AI project, described in *Appendix B.1, page 19, line 1033-1036*. The text from the Appendix is quoted below:
> > “In addition to our proposed new imagination, another example of the new imagination of AI is the Indigenous AI project. Named Abundant Intelligences, this new research agenda of AI is based on Indigenous knowledge systems and relational ethics. It reimagines how AI technologies can flourish the Indigenous communities and how AI development can be guided towards a more humane future (Lewis et al., 2024; 2020).”
>
> Based on your suggestion, in our revision, in addition to our existing example in Section 4.3 of the case study, we will refer to and highlight other examples of more real-world examples in the Conclusion section, to showcase other ongoing initiatives and possibilities of constructing the diverse and ethical imaginations of AI. The examples that we will cite are listed below:
>
> * Revisit and highlight the various AI ethics communities mentioned in the Introduction, including AI for (Social) Good, AI for Science, and FaccT (fairness, accountability, and transparency, as brought up by Reviewer UbQM)
> * The Indigenous AI project mentioned above
> * [The First Languages AI Reality (FLAIR) initiative](https://mila.quebec/en/ai4humanity/applied-projects/first-languages-ai-reality) that revitalizes Indigenous language with AI
> * [Mozilla Common Voice](https://foundation.mozilla.org/en/common-voice/): a diverse open voice dataset project for voice recognition
> * [Fairly Trained](https://www.fairlytrained.org/): a non-profit that certifies fair training data use in generative AI to get a fairer deal for human creators

---

### Decision · Program_Chairs · 2025-04-27

**Decision:**

Reject

**Comment:**

The paper was reviewed by four expert reviewers. The paper's strengths and weaknesses are summarized as following.

The chief strengths includes its clear focus on ethical and societal shortcomings of AI and introduction to a new framework ``AI for just work'' which prioritizes workers' rights, ethical values, and real-world grounding. The paper is well-supported with sufficient references and provides an in-depth discussions at the intersection of AI and ethics. The discussion is likely to inspire debate about AI's role in labor, fairness, and ethical decision-making. Also, there are authors' good intentions in highlighting the potential risks of AI and advocating for a broader discussion on the motivations behind AI development.

On the other hand, there are serious weaknesses in the paper. First, the paper is lacking in concrete technical methodologies and real-world examples. Second, the paper's position is rather unclear - it presents more like a proposal of a new method rather than a clear stance on a particular research priority or recommendation for changes. It is also lacking in empirical or statistical evidence supporting the core claim that the ML community is a ``mono-culture shaped by a small and largely homogeneous group of people.'' Lastly, the example of medical image synthesis is confusing, as researchers are more focuses on AI-assisted diagnosis, which is a different problem.

The authors' rebuttal was read and considered by the reviewers. While it addressed some of the original concerns from the reviewers, there remained concerns about unclear position and insufficient empirical evidences. These issues are critical for a position paper. The AC agreed with the reviewers' opinions and recommended rejection of the work.